# GENERATING MOLECULAR CONFORMER FIELDS

## ABSTRACT

In this paper we tackle the problem of generating conformers of a molecule in 3D space given its molecular graph. We parameterize these conformers as continuous functions that map elements from the molecular graph to points in 3D space. We then formulate the problem of learning to generate conformers as learning a distribution over these functions using a diffusion generative model, called Molecular Conformer Fields (MCF). Our approach is simple and scalable, and achieves state-of-the-art performance on challenging molecular conformer generation benchmarks while making no assumptions about the explicit structure of molecules (*e.g.* modeling torsional angles). MCF represents an advance in extending diffusion models to handle complex scientific problems in a conceptually simple, scalable and effective manner.

## 1 INTRODUCTION

In this paper we tackle the problem of Molecular Conformer Generation, *i.e.* predicting the diverse low-energy three-dimensional conformers of molecules, relying solely on their molecular graphs as illustrated in Fig. 1. Molecular Conformer Generation is a fundamental problem in computational drug discovery and chemo-informatics, where understanding the intricate interactions between molecular and protein structures in 3D space is critical, affecting aspects such as charge distribution, potential energy, etc. (Batzner et al., 2022). The core challenge associated with conformer generation springs from the vast complexity of the 3D structure space, encompassing factors such as bond lengths and torsional angles. Despite the molecular graph dictating potential 3D conformers through specific constraints, such as bond types and spatial arrangements determined by chiral centers, the conformational space experiences exponential growth with the expansion of the graph size and the number of rotatable bonds (Axelrod & Gomez-Bombarelli, 2022). This complicates brute force approaches, making them virtually unfeasible for even moderately small molecules.

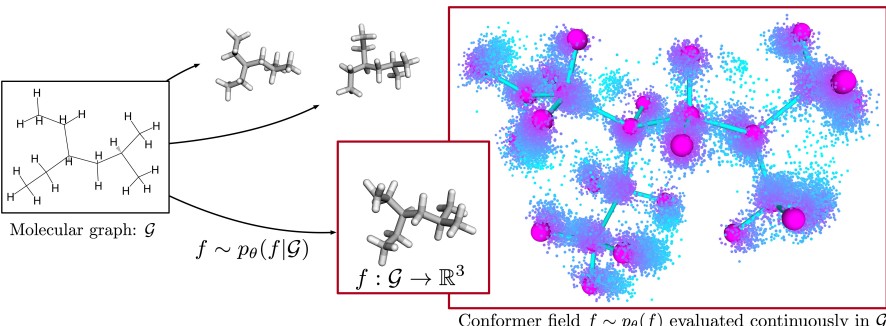

Figure 1: We formulate the molecular conformer generation problem as learning a distribution of functions over graphs. Each conformer is a field (*e.g.* a function) $f$ that can map elements from a particular molecular graph $\mathcal{G}$ to points in 3D space, $f : \mathcal{G} \to \mathbb{R}^3$. We call these *conformer fields* as they can be continuously evaluated for any point in $\mathcal{G}$. We visualize an example of example of this conformer field where points are colored based on the distance to their closest atom (see Sect. 5.3 for details).

Systematic methods, like OMEGA (Hawkins et al., 2010), offer rapid processing through rule-based generators and curated torsion templates. Despite their efficiency, these models typically fail on

complex molecules, as they often overlook global interactions and are tricky to extend to inputs like transition states or open-shell molecules. Classic stochastic methods, like molecular dynamics (MD) and Markov chain Monte Carlo (MCMC), rely on extensively exploring the energy landscape to find low-energy conformers. Such techniques suffer from sampling inefficiency for large molecules and struggle to generate diverse representative conformers (Hawkins, 2017; Wilson et al., 1991; Grebner et al., 2011). In the domain of learning-based approaches, several works have looked at conformer generation problems through the lens of probabilistic modeling, using either normalizing flows (Xu et al., 2021a) or diffusion models (Xu et al., 2022; Jing et al., 2022). These approaches tend to use equivariant network architectures to deal with molecular graphs (Xu et al., 2022) or model domain-specific factors like torsional angles (Ganea et al., 2021; Jing et al., 2022). However, explicitly enforcing these domain-specific inductive biases can sometimes come at a cost. For example, Torsional Diffusion relies on rule-based methods to find rotatable bonds which may fail especially for complex molecules. Also, the quality of generated conformers are adhered to the non-differentiable cheminformatic methods used to predict local substructures. On the other hand, recent works have proposed domain-agnostic approaches for generative modeling of data in function space (Du et al., 2021; Dupont et al., 2022b;a; Zhuang et al., 2023) obtaining great performance. As an example, in Zhuang et al. (2023) the authors use a diffusion model to learn a distribution over fields $f$, showing great results on different data domains like images (*i.e.* $f : \mathbb{R}^2 \to \mathbb{R}^3$) or 3D geometry (*i.e.* $f : \mathbb{R}^3 \to \mathbb{R}^1$), where the domain of the function $\mathbb{R}^n$ is fixed across functions. However, dealing with fields defined on different domains (*e.g.* different molecular graphs, as in molecular conformer generation) still remains an open problem.

To address these issues, we present Molecular Conformer Fields (MCF), an approach to learn generative models of molecular conformers. We interpret conformers as fields/functions (we use both terms exchangeably) on graphs that map elements in the graph $\mathcal{G}_i$ to points in $\mathbb{R}^3$, $f_i : \mathcal{G}_i \to \mathbb{R}^3$, which we define as *conformer fields*. MCF present a generalization over previous work on generative models in function space (Du et al., 2021; Dupont et al., 2022b;a; Zhuang et al., 2023) to deal with functions $f_i$ defined on different graphs $\mathcal{G}_i$ while also generalizing those approaches to dealing with intrinsic coordinate systems. MCF outperforms recent conformer generation benchmarks while being conceptually simpler to understand and efficient to scale. In addition, successfully tackling the problem of conformer generation with a general approach like MCF shows that it is a strong method that can generalize to other scientific disciplines with little effort. In addition, MCF can be a stepping stone towards all-purpose diffusion generative models.

Our contributions can be summarized as follows:

- We provide a simple yet effective approach for molecule conformer generation that achieves state-of-the-art performance on standard benchmarks.
- Our approach directly predicts the 3D position of atoms as opposed to torsional angles or other domain-specific variables, providing a simple and scalable training recipe.
- We provide an extensive ablation study to understand what are the critical factors that affect the performance of our proposed MCF in molecular conformer generation.

## 2    RELATED WORK

Recent works have tackled the problem of conformer generation using learning-based generative models. Simm & Hernández-Lobato (2019) and Xu et al. (2021b) develop two-stage methods which first generate interatomic distances following VAE framework and then predict conformers based on the distances. Guan et al. (2021) propose neural energy minimization to optimize low-quality conformers. In Xu et al. (2021a), a normalizing flow approach is proposed as an alternative to VAEs. To avoid the accumulative errors from two-stage generation, Shi et al. (2021) implement score-based generative model to directly model the gradient of logarithm density of atomic coordinates. In GeoDiff (Xu et al., 2022), a diffusion model is used which focuses on crafting equivariant forward and backward processes with equivariant graph neural networks. In GeoMol (Ganea et al., 2021), the authors first predict 1-hop local structures and then propose a regression objective coupled with an Optimal Transport loss to predict the torsional angles that assemble substructures of a molecule. Following this, Torsional Diffusion (Jing et al., 2022) proposed a diffusion model on the torsional angles of the bonds rather than a regression model used in Ganea et al. (2021).

Our approach extends recent efforts in generative models for continuous functions in Euclidean space (Zhuang et al., 2023; Dupont et al., 2022b;a; Du et al., 2021), to functions defined over graphs (*e.g.* chemical structure of molecules). The term Implicit Neural Representation (INR) is used in these works to denote a parameterization of a single function (*e.g.* a single image in 2D) using a neural network that maps the function's inputs (*i.e.* pixel coordinates) to its outputs (*i.e.* RGB values). Different approaches have been proposed to learn distributions over fields in Euclidean space; GASP (Dupont et al., 2022b) leverages a GAN whose generator produces field data whereas a point cloud discriminator operates on discretized data and aims to differentiate real and generated functions. Two-stage approaches (Dupont et al., 2022a; Du et al., 2021) adopt a latent field parameterization (Park et al., 2019) where functions are parameterized via a hyper-network (Ha et al., 2017) and a generative model is learnt on the latent or INR representations. In addition, our approach also relates to recent work focusing on fitting a function (*e.g.* learning an INR) on a manifold using an intrinsic coordinate system (Koestler et al., 2022; Grattarola & Vandergheynst, 2022), and generalizes it to the problem of learning a probabilistic model over multiple functions defined on different manifolds/graphs. Intrinsic coordinate systems have also been used in Graph Transformers to tackle supervised learning tasks (Maskey et al., 2022; Sharp et al., 2022; He et al., 2022; Dwivedi et al., 2020).

Recent strides in the domain of protein folding dynamics have witnessed revolutionary progress, with modern methodologies capable of predicting crystallized 3D structures solely from amino-acid sequences using auto-regressive models like AlphaFold (Jumper et al., 2021). However, transferring these approaches seamlessly to general molecular data is fraught with challenges. Molecules present a unique set of complexities, manifested in their highly branched graphs, varying bond types, and chiral information, aspects that make the direct application of protein folding strategies to molecular data a challenging endeavor.

## 3 PRELIMINARIES

### 3.1 DIFFUSION PROBABILISTIC FIELDS

Diffusion Probabilistic Fields (DPF) (Zhuang et al., 2023) belong to the broad family of latent variable models (Everett, 2013) and can be consider a generalization of DDPMs (Ho et al., 2020) to deal with continuous functions $f : \mathbf{M} \to Y$ which are infinite dimensional. Conceptually speaking, DPF (Zhuang et al., 2023) parametrizes functions $f$ with a set of context pairs containing input-outputs to the function. Using these context pairs as input to DPF, the model is trained to denoise any continous query coordinate (*e.g.* query pairs) in the domain of the function at timestep $t$. In order to learn a parametric distribution over functions $p_\theta(f_0)$ from an empirical distribution of functions s $q(f_0)$, DPF reverses a diffusion Markov Chain that generates function latents $f_{1:T}$ by gradually adding Gaussian noise to (context) input-output pairs randomly drawn from $f \sim q(f_0)$ for $T$ time-steps as follows: $q(f_t|f_{t-1}) := \mathcal{N}(f_{t-1}; \sqrt{\bar{\alpha}_t}f_0, (1 - \bar{\alpha}_t)\mathbf{I})$. Here, $\bar{\alpha}_t$ is the cumulative product of fixed variances with a handcrafted scheduling up to time-step $t$. $q(f_t|f_{t-1})$ denotes the conditional distribution on the output space (which follows a Gaussian) since the domain of the function $\mathcal{M}$ never changes, only the values assigned to it. DPF (Zhuang et al., 2023) follows the training recipe in Ho et al. (2020) in which: i) The forward process adopts sampling in closed form. ii) reversing the diffusion process is equivalent to learning a sequence of denoising (or score) networks $\epsilon_\theta$, with tied weights. Reparameterizing the forward process as $f_t = \sqrt{\bar{\alpha}_t}f_0 + \sqrt{1 - \bar{\alpha}_t}\epsilon$ results in the "simple" DDPM loss: $\mathbb{E}_{t \sim [0,T], f_0 \sim q(f_0), \epsilon \sim \mathcal{N}(0,\mathbf{I})}\left[\|\epsilon - \epsilon_\theta(\sqrt{\bar{\alpha}_t}f_0 + \sqrt{1 - \bar{\alpha}_t}\epsilon, t)\|^2\right]$, which makes learning of the data distribution $p_\theta(f_0)$ both efficient and scalable. At inference time, DPF computes $f_0 \sim p_\theta(f_0)$ via ancestral sampling (Zhuang et al., 2023). Concretely, DPF starts by sampling dense query coordinates and assigning a Gaussian value to them $f_T \sim \mathcal{N}(\mathbf{0}, \mathbf{I})$. Then, it iteratively applies the score network $\epsilon_\theta$ to denoise $f_T$, thus reversing the diffusion Markov Chain to obtain $f_0$. In practice, DPFs have obtained amazing results for signals living in an Euclidean geometry. *However, the extension to functions defined on graphs remains an open problem.*

### 3.2 CONFORMERS AS FUNCTIONS ON GRAPHS

Following the setting in previous work (Xu et al., 2022; Ganea et al., 2021; Jing et al., 2022) a molecule with $n$ atoms is represented as an undirected graph $\mathcal{G} = \langle \mathcal{V}, \mathcal{E} \rangle$, where $\mathcal{V} = \{v_i\}_{i=1}^n$ is the set of vertices representing atoms and $\mathcal{E} = \{e_{ij}|(i,j) \subseteq |\mathcal{V}| \times |\mathcal{V}|\}$ is the set of edges representing

inter-atomic bonds. In this paper, we parameterize a molecule's conformer as a function $f : \mathcal{G} \to \mathbb{R}^3$ that takes atoms in the molecular graph $\mathcal{G}$ and maps them to 3D space, we call this function a *conformer field*. The training set is composed of conformer fields $f_i : \mathcal{G}_i \to \mathbb{R}^3$, that each maps atoms of a different molecule $\mathcal{G}_i$ to a 3D point. We then formulate the task of conformer generation as learning a prior over a training set of conformer fields. We drop the subscript $i$ in the remainder of the text for notation simplicity.

We learn a denoising diffusion generative model (Ho et al., 2020) over conformer fields $f$. In particular, given conformer field samples $f_0 \sim q(f_0)$ the forward process takes the form of a Markov Chain with progressively increasing Gaussian noise: $q(f_{1:T}|f_0) = \prod_{t=1}^T q(f_t|f_{t-1})$, $q(f_t|f_{t-1}) :=$ $\mathcal{N}(f_{t-1}; \sqrt{\bar{\alpha}_t} f_0, (1 - \bar{\alpha}_t)\mathbf{I})$. It should be noted that $q(f_t|f_{t-1})$ represents the conditional Gaussian distribution on the output space of the field instead of the field itself, as the domain of the function $\mathcal{M}$ never changes, only the values assigned to it. We train MCF using the denoising objective function in (Ho et al., 2020): $\mathbb{E}_{t \sim [0,T], f_0 \sim q(f_0), \epsilon \sim \mathcal{N}(0,\mathbf{I})} \left[ \| \epsilon - \epsilon_\theta(\sqrt{\bar{\alpha}_t} f_0 + \sqrt{1 - \bar{\alpha}_t}\epsilon, t) \|^2 \right]$.

### 3.3 EQUIVARIANCE IN CONFORMER GENERATION

Equivariance has become an important topic of study in generative models (Abbott et al., 2023; 2022; Kanwar et al., 2020). In particular, enforcing equivariance as an explicit inductive bias in neural networks can often lead to improved generalization (Köhler et al., 2020) by constraining the space of functions that can be represented by a model. On the other hand, recent literature shows that models that can learn these symmetries from data rather than explicitly enforcing them (*e.g.* transformers vs CNNs) tend to perform better as they are more amenable to optimization (Bai et al., 2021).

Equivariance also plays an interesting role in conformer generation. On one hand, it is important when training likelihood models of conformers, as the likelihood of a conformer is invariant to roto-translations (Köhler et al., 2020). On the other hand, when training models to generate conformers given a molecular graph, explicitly baking roto-translation equivariance might not be as necessary. This is because the intrinsic structure of the conformer encodes far more information about its properties than the extrinsic coordinate system (eg. rotation and translation) in which the conformer is generated (Ruddigkeit et al., 2012). In addition, recent approaches for learning simulations on graphs (Sanchez-Gonzalez et al., 2020) or pre-training models for molecular prediction tasks (Zaidi et al., 2022) have relied on non-equivariant architectures.

We follow this trend and empirically show that explicitly enforcing roto-translation equivariance is not a strong requirement for generalization. Furthermore, we also show that approaches that do not explicitly enforce roto-translation equivariance (like ours) can match or outperform approaches that do.

## 4 METHOD

MCF is a diffusion generative model that captures distributions over conformer fields. We are given observations in the form of an empirical distribution $f_0 \sim q(f_0)$ over fields where a field $f_0 : \mathcal{G} \to \mathbb{R}^3$ maps vertices $v \in \mathcal{G}$ on a molecular graph $\mathcal{G}$ to 3D space $\mathbb{R}^3$. As a result, latent variables $f_{1:T}$ are also functions on graphs that can be continuously evaluated.

To tackle the problem of learning a diffusion generative model over conformer fields we extend the recipe in DPF (Zhuang et al., 2023), generalizing from fields defined in ambient Euclidean space to functions on graphs (*e.g.* conformer fields). In order to do this, we compute the $k$ leading eigenvectors of the normalized graph Laplacian $\Delta_\mathcal{G}$ (Maskey et al., 2022; Sharp et al., 2022) as positional encoding for points in the graph. The eigen-decomposition of the normalized graph Laplacian can be computed efficiently using sparse eigen-problem solvers (Hernandez et al., 2009) and only needs to be computed once before training. We use the term $\varphi(v) = \sqrt{n}[\varphi_1(v), \varphi_2(v), \ldots, \varphi_k(v)] \in \mathbb{R}^k$ to denote the normalized Laplacian eigenvector representation of a vertex $v \in \mathcal{G}$.

We adopt an explicit field parametrization where a field is characterized by uniformly sampling a set of vertex-signal pairs $\{(\varphi(v_c), \boldsymbol{y}_{(c,0)})\}$, $v_c \in \mathcal{G}$, $\boldsymbol{y}_{(c,0)} \in \mathbb{R}^3$, which is denoted as *context set*. We row-wise stack the context set and refer to the resulting matrix via $\mathbf{C}_0 = [\varphi(\mathcal{V}_c), \mathbf{Y}_{(c,0)}]$. Here, $\varphi(\mathcal{V}_c)$ denotes the Laplacian eigenvector representation context vertices and $\mathbf{Y}_{(c,0)}$ denotes the 3D

---

**Algorithm 1** Training

1: $\Delta_{\mathcal{G}} \varphi_i = \varphi_i \lambda_i$ // Compute Laplacian eigenvectors
2: **repeat**
3:   $(\mathbf{C}_0, \mathbf{Q}_0) \sim \text{Uniform}(q(f_0))$
4:   $t \sim \text{Uniform}(\{1, \ldots, T\})$
5:   $\epsilon_c \sim \mathcal{N}(\mathbf{0}, \mathbf{I}), \epsilon_q \sim \mathcal{N}(\mathbf{0}, \mathbf{I})$
6:   $\mathbf{C}_t = [\varphi(\mathcal{V}_c), \sqrt{\bar{\alpha}_t} \mathbf{Y}_{(c,0)} + \sqrt{1 - \bar{\alpha}_t} \epsilon_c]$
7:   $\mathbf{Q}_t = [\varphi(\mathcal{V}_q), \sqrt{\bar{\alpha}_t} \mathbf{Y}_{(q,0)} + \sqrt{1 - \bar{\alpha}_t} \epsilon_q]$
8:   Take gradient descent step on
      $\nabla_\theta \|\epsilon_q - \epsilon_\theta(\mathbf{C}_t, t, \mathbf{Q}_t)\|^2$
9: **until** converged

Figure 2: **Left:** MCF training algorithm. **Right**: Visual depiction of a training iteration for a conformer field. See Sect. 4 for definitions (.

position of context vertices at time $t = 0$. We define the forward process for the context set by diffusing the 3D positions and keeping Laplacian eigenvectors fixed:

$$\mathbf{C}_t = [\varphi(\mathcal{V}_c), \mathbf{Y}_{(c,t)} = \sqrt{\bar{\alpha}_t} \mathbf{Y}_{(c,0)} + \sqrt{1 - \bar{\alpha}_t} \epsilon_c], \tag{1}$$

where $\epsilon_c \sim \mathcal{N}(\mathbf{0}, \mathbf{I})$ is a noise vector of the appropriate size. We now turn to the task of formulating a score network for fields. The score network needs to take as input the context set (*i.e.* the field parametrization), and needs to accept being evaluated continuously in $\mathcal{G}$. We do this by sampling a *query set* of vertex-signal pairs $\{\varphi(v_q), \boldsymbol{y}_{(q,0)}\}$. Equivalently to the context set, we row-wise stack query pairs and denote the resulting matrix as $\mathbf{Q}_0 = [\varphi(\mathcal{V}_q), \mathbf{Y}_{(q,0)}]$. Note that the forward diffusion process is equivalently defined for both context and query sets:

$$\mathbf{Q}_t = [\varphi(\mathcal{V}_q), \mathbf{Y}_{(q,t)} = \sqrt{\bar{\alpha}_t} \mathbf{Y}_{(q,0)} + \sqrt{1 - \bar{\alpha}_t} \epsilon_q], \tag{2}$$

where $\epsilon_q \sim \mathcal{N}(\mathbf{0}, \mathbf{I})$ is a noise vector of the appropriate size. The underlying field is solely defined by the context set, and the query set are the function evaluations to be de-noised. The resulting *score field* model is formulated as follows, $\hat{\epsilon}_q = \epsilon_\theta(\mathbf{C}_t, t, \mathbf{Q}_t)$.

Using the explicit field characterization and the score field network, we obtain the training and inference procedures in Alg. 1 and Alg. 2, respectively, which are accompanied by illustrative examples of sampling a field encoding a Gaussian mixture model over the manifold (*i.e.* the bunny). For training, we uniformly sample context and query sets from $f_0 \sim \text{Uniform}(q(f_0))$ and only corrupt their signal using the forward process in Eq. equation 1 and Eq. equation 2. We train the score field network $\epsilon_\theta$ to denoise the signal portion of the query set, given the context set. During sampling, to generate a conformer fields $f_0 \sim p_\theta(f_0)$ we first define a query set $\mathbf{Q}_T = [\varphi(\mathcal{V}_q), \mathbf{Y}_{(q,T)} \sim \mathcal{N}(\mathbf{0}, \mathbf{I})]$ of random values to be de-noised. We set the context set to be a random subset of the query set. We use the context set to denoise the query set and follow ancestral sampling as in the vanilla DDPM (Ho et al., 2020). Note that during inference the eigen-function representation $\varphi(v)$ of the context and query sets does not change, only their corresponding signal value (*e.g.* their 3D position).

### 4.1 SCORE FIELD NETWORK $\epsilon_\theta$

In MCF, the score field's design space covers all architectures that can process irregularly sampled data, such as Transformers (Vaswani et al., 2017) and their corresponding Graph counterparts (Maskey et al., 2022; Sharp et al., 2022; He et al., 2022; Dwivedi et al., 2020) which have recently gained popularity in the supervised learning setting. The score field network $\epsilon_\theta$ is primarily implemented using PerceiverIO (Jaegle et al., 2022), an effective transformer encoder-decoder architecture. The PerceiverIO was chosen due to its efficiency in managing large numbers of elements in the context and query sets, as well as its natural ability to encode interactions between these sets using attention.

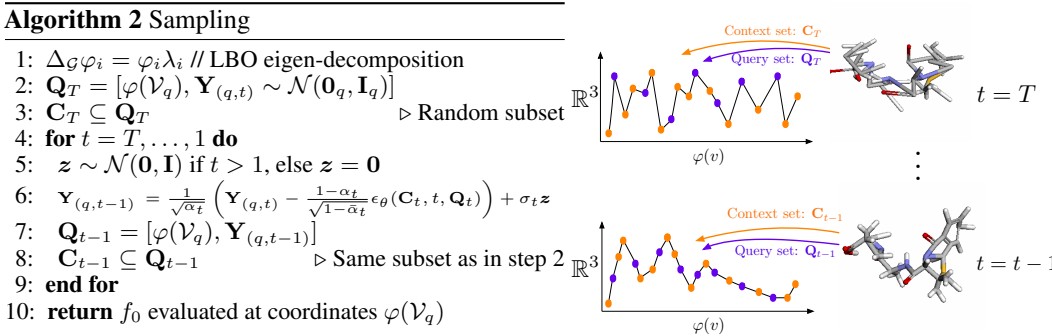

**Algorithm 2** Sampling

1: $\Delta_{\mathcal{G}}\varphi_i = \varphi_i\lambda_i$ // LBO eigen-decomposition
2: $\mathbf{Q}_T = [\varphi(\mathcal{V}_q), \mathbf{Y}_{(q,t)} \sim \mathcal{N}(\mathbf{0}_q, \mathbf{I}_q)]$
3: $\mathbf{C}_T \subseteq \mathbf{Q}_T$ ▷ Random subset
4: **for** $t = T, \ldots, 1$ **do**
5:   $\mathbf{z} \sim \mathcal{N}(\mathbf{0}, \mathbf{I})$ if $t > 1$, else $\mathbf{z} = \mathbf{0}$
6:   $\mathbf{Y}_{(q,t-1)} = \frac{1}{\sqrt{\alpha_t}}\left(\mathbf{Y}_{(q,t)} - \frac{1-\alpha_t}{\sqrt{1-\bar{\alpha}_t}}\epsilon_\theta(\mathbf{C}_t, t, \mathbf{Q}_t)\right) + \sigma_t \mathbf{z}$
7:   $\mathbf{Q}_{t-1} = [\varphi(\mathcal{V}_q), \mathbf{Y}_{(q,t-1)}]$
8:   $\mathbf{C}_{t-1} \subseteq \mathbf{Q}_{t-1}$ ▷ Same subset as in step 2
9: **end for**
10: **return** $f_0$ evaluated at coordinates $\varphi(\mathcal{V}_q)$

Figure 3: **Left:** MCF sampling algorithm. **Right**: Visual depiction of the sampling process of a conformer field.

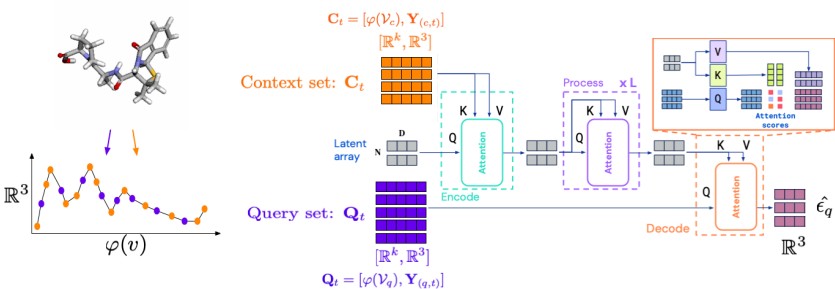

Figure 4: Interaction between context and query pairs in the PerceiverIO architecture. Context pairs $\mathbf{C}_t$ attend to a latent array of learnable parameters via cross attention. The latent array then goes through several self attention blocks. Finally, the query pairs $\mathbf{Q}_t$ cross-attend to the latent array to produce the final noise prediction $\hat{\epsilon}_q$.

Figure 4 demonstrates how these sets are used within the PerceiverIO architecture. To elaborate, the encoder maps the context set into latent arrays (*i.e.* a group of learnable vectors) through a cross-attention layer, while the decoder does the same for query set. For a more detailed analysis of the PerceiverIO architecture refer to Jaegle et al. (2022).The time-step $t$ is incorporated into the score computation by concatenating a positional embedding representation of $t$ to the context and query sets.

## 5 EXPERIMENTS

We use two popular datasets: GEOM-QM9 (Ruddigkeit et al., 2012) and GEOM-DRUGS (Ruddigkeit et al., 2012). Datasets are preprocessed as described in GeoMol (Ganea et al., 2021). We split them randomly based on molecules into train/validation/test (80%/10%/10%). At the end, for each dataset, we sample 1000 random test molecules as the final test set. Thus, the splits contain 106586/13323/1000 and 243473/30433/1000 molecules for GEOM-QM9 and GEOM-DRUGS, resp. We follow the exact same training splits for all baselines. In the Appendix A.3 we provide additional experiments that validate the design choices for the score network architecture, as well as empirically validating the chemical properties of generated conformers.

### 5.1 GEOM-QM9

Following the standard setting for molecule conformer prediction we use the GEOM-QM9 dataset (Ruddigkeit et al., 2012) which contains $\sim 130K$ molecules ranging from 3 to 29 atoms. We report our results in Tab. 1 and compare with CGCF (Xu et al., 2021a), GeoDiff (Xu et al., 2022), GeoMol

(Ganea et al., 2021) and Torsional Diffusion (Jing et al., 2022). Note that both GeoMol (Ganea et al., 2021) and Torsional Diffusion (Jing et al., 2022) make strong assumptions about the geometric structure of molecules and model domain-specific characteristics (*i.e.* torsional angles of rotatable bonds). In contrast, MCF simply models the distribution of 3D coordinates of atoms without making any assumptions about the underlying structure. Finally we report the same metrics as Torsional Diff. (Jing et al., 2022) to compare the generated and ground truth conformer ensembles: Average Minimum RMSD (AMR) and Coverage. These metrics are reported both for precision, measuring the accuracy of the generated conformers, and recall, which measures how well the generated ensemble covers the ground-truth ensemble (details about metrics can be found in Appendix A.2.4). We generate $2K$ conformers for a molecule with $K$ ground truth conformers.

We report results in Tab. 1, showing that MCF outperforms previous approaches by a substantial margin. In addition, it is important to note that MCF is a general approach for learning functions on graphs that does not make any assumptions about the intrinsic geometric factors important in conformers like torsional angles. This makes MCF simpler to implement and applicable to other settings in which intrinsic geometric factors are not known or expensive to compute.

| | Recall | | | | Precision | | | |
|---|---|---|---|---|---|---|---|---|
| | Coverage ↑ | | AMR ↓ | | Coverage ↑ | | AMR ↓ | |
| | mean | median | mean | median | mean | median | mean | median |
| CGCF | 69.47 | 96.15 | 0.425 | 0.374 | 38.20 | 33.33 | 0.711 | 0.695 |
| GeoDiff | 76.50 | **100.00** | 0.297 | 0.229 | 50.00 | 33.50 | 0.524 | 0.510 |
| GeoMol | 91.50 | **100.00** | 0.225 | 0.193 | 87.60 | **100.00** | 0.270 | 0.241 |
| Torsional Diff. | 92.80 | **100.00** | 0.178 | 0.147 | 92.70 | **100.00** | 0.221 | 0.195 |
| MCF (ours) | **96.57** | **100.00** | **0.107** | **0.072** | **94.60** | **100.00** | **0.130** | **0.084** |

Table 1: Molecule conformer generation results on GEOM-QM9. MCF obtains better results than the state-of-the-art Torsional Diffusion (Jing et al., 2022), without making any explicit assumptions about the geometric structure of molecules (*i.e.* without modeling torsional angles).

## 5.2 GEOM-DRUGS

To test the capacity of MCF to deal with larger molecules we also report experiments on GEOM-DRUGS, the largest and most pharmaceutically relevant part of the GEOM dataset (Axelrod & Gomez-Bombarelli, 2022) — consisting of 304k drug-like molecules (average 44 atoms). We report our results in Tab. 2 and compare with GeoDiff (Xu et al., 2022), GeoMol (Ganea et al., 2021) and Torsional Diffusion (Jing et al., 2022). Note again that both GeoMol (Ganea et al., 2021) and Torsional Diffusion (Jing et al., 2022) make strong assumptions about the geometric structure of molecules and model domain-specific characteristics like torsional angles of bonds. In contrast, MCF simply models the distribution of 3D coordinates of atoms without making any assumptions about the underlying structure.

Results on Tab. 2 are where we see how MCF outperforms strong baseline approaches by substantial margins. MCF achieves better performance than previous state-of-the-art model Torsional Diffusion by approximately 10% on both recall and precision. This indicates that our proposed MCF not only generates high-quality conformers that are close with ground truth but also covers a wide variety of conformers in the ground truth. In addition, it is important to note that MCF does not make any assumptions about the intrinsic geometric factors important in conformers like torsional angles and thus is simpler to train (*e.g.* it does not require a local substructure prediction model).

In addition, in Fig. 5 we show a breakdown of the performance on GEOM-DRUGS of MCF vs. Torsional diffusion (Jing et al., 2022) as a function of the threshold distance, as well as a function of the number of atoms in molecules. MCF outperforms Torsional Diffusion across the full spectrum of thresholds in both recall and precision. When looking at the break-down AMR on different number of atoms in Fig. 5(c), MCF also demonstrates its superior performance in most cases. It is indicated that MCF better captures the fine intrinsic geometric structure of conformers.

| | Recall | | | | Precision | | | |
|---|---|---|---|---|---|---|---|---|
| | Coverage ↑ | | AMR ↓ | | Coverage ↑ | | AMR ↓ | |
| | mean | median | mean | median | mean | median | mean | median |
| GeoDiff | 42.10 | 37.80 | 0.835 | 0.809 | 24.90 | 14.50 | 1.136 | 1.090 |
| GeoMol | 44.60 | 41.40 | 0.875 | 0.834 | 43.00 | 36.40 | 0.928 | 0.841 |
| Torsional Diff. | 72.70 | 80.00 | 0.582 | 0.565 | 55.20 | 56.90 | 0.778 | 0.729 |
| MCF (ours) | **81.62** | **89.22** | **0.468** | **0.438** | **61.63** | **62.50** | **0.705** | **0.650** |

Table 2: Molecule conformer generation results on GEOM-DRUGS. MCF obtains comparable or better results than the state-of-the-art Torsional Diffusion Jing et al. (2022), without making any explicit assumptions about the geometric structure of molecules (*i.e.* without modeling torsional angles).

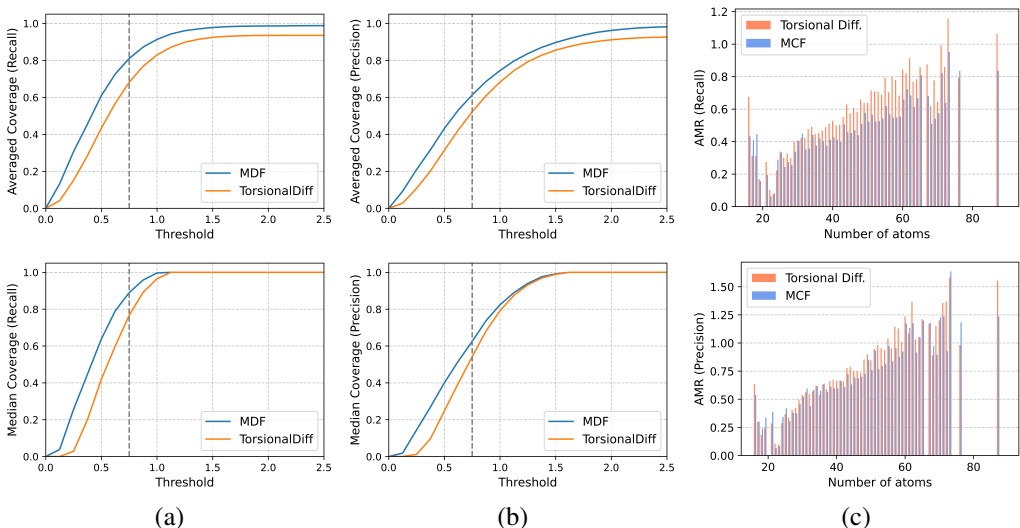

(a)  (b)  (c)

Figure 5: (a) Recall coverage metric as a function of the threshold distance. MCF outperforms Torsional Diffusion across the full spectrum of thresholds. (b) Precision coverage as a function of the threshold distance. Still MCF outperforms Torsional Diffusion across the full spectrum of thresholds. (c) Minimum averaged RMSD (lower is better) of recall and precision as a function of the number of atoms in molecules.

### 5.2.1 GENERALIZATION TO GEOM-XL

We now turn to the task of evaluating how well a model trained on GEOM-DRUGS transfers to unseen molecules with large numbers of atoms. As proposed in Torsional Diff. (Jing et al., 2022) we use the GEOM-XL dataset, which is a subset of GEOM-MoleculeNet that contains all species with more than 100 atoms, which is a total of 102 molecules. Note that this evaluation not only tests the capacity of models to generalize to molecules with large number of atoms but also serves as an out-of-distribution generalization experiment.

In Tab. 3 we report AMR for both precision and recall and compare with GeoDiff (Xu et al., 2022), GeoMol (Ganea et al., 2021) and Torsional Diffusion (Jing et al., 2022). In particular, when taking the numbers directly from (Jing et al., 2022), MCF achieves better or comparable performance than Torsional Diffusion. Further, in running the checkpoint provided by Torsional Diffusion and following their validation process we found that 25 molecules failed to be generated, this is due to the fact that Torsional Diffusion generates torsional angles conditioned on the molecular graph $\mathcal{G}$ and the local structures obtained from RDKit. In some cases, RDKit can fail to find local structures and Torsional Diffusion cannot generate conformers in this case. In our experiments with the same 77 molecules in GEOM-XL from our replica, MCF surpasses Torsional Diffusion by a large margin. The results highlight the generalizability of MCF to large and complex molecules, which may shed a light on pre-training molecular conformer generation model.

| | AMR-P ↓ | | AMR-R ↓ | | # mols |
|---|---|---|---|---|---|
| | mean | median | mean | median | |
| GeoDiff | 2.92 | 2.62 | 3.35 | 3.15 | - |
| GeoMol | 2.47 | 2.39 | 3.30 | 3.14 | - |
| Torsional Diff. (Jing et al., 2022) | 2.05 | 1.86 | **2.94** | 2.78 | - |
| MCF | **2.04** | **1.72** | 2.97 | **2.51** | 102 |
| Torsional Diff. (our eval) | 1.93 | 1.86 | 2.84 | 2.71 | 77 |
| MCF | **1.77** | **1.59** | **2.62** | **2.37** | 77 |

Table 3: Generalization results on GEOM-XL. MCF obtains comparable results to Torsional Diffusion Jing et al. (2022), without making any explicit assumptions about the geometric structure of molecules.

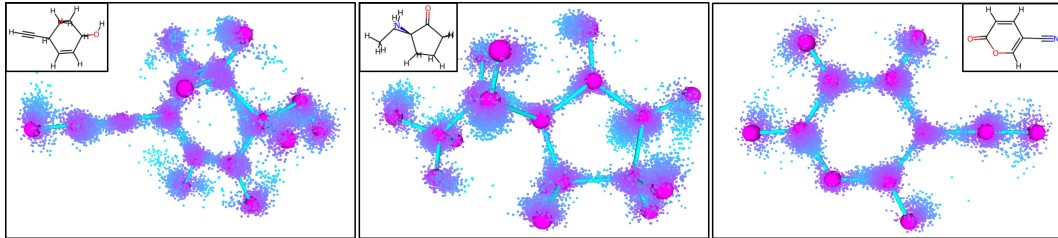

Figure 6: Continuously evaluating generated conformer fields for different molecules in GEOM-QM9.

## 5.3 CONTINUOUS CONFORMERS

Molecules are ubiquitously represented as graphs $\mathcal{G}$ where vertices $\mathcal{V}$ of the graph represents atoms and their properties and edges represents bonds that encode interactions between atoms. These bonds encode electron interactions between atoms but also serve to describe the nature and characteristics of such interactions (*i.e.* covalent bonds, ionic bonds, metallic bonds, etc, ). In particular, the molecule's conformer is generated only for discrete atoms (*e.g.* vertices in $\mathcal{G}$). However, since MCF encodes continuous conformer fields it can be *continuously evaluated* in $\mathcal{G}$. For example, we can evaluate the sampled conformer fields for points along bonds. In order to do this, for a point $p$ in a bond connecting atoms $(v_i, v_j)$ we linearly interpolate the Laplacian eigenvector representation of it's endpoints $\varphi(p) = \alpha\varphi(v_i) + (1 - \alpha)\varphi(v_j)$, we then feed this interpolated Laplacian eigenvector into the model to sample its 3D position in the conformer field. We visualize results in Fig. 1 and 6. We generated this visualizations an MCF model trained on GEOM-QM9 without atom features. Note that while MCF is never trained on points along molecular bonds, it manages to generate plausible 3D positions for such points.

This flexibility to evaluate conformers continuously in $\mathcal{G}$ opens a realm of possibilities. For example, at this subatomic scales, the paths that electrons take are not well-defined tracks, but rather regions of space where they are most likely to be found, represented by probability density functions. These electron clouds form the bonds between atoms, and their shapes can vary quite a bit depending on the bond type, being sometimes symmetrical and other times quite complex and diffuse. MCF enables the possibility of using additional training data from Quantum Monte Carlo methods (Nightingale & Umrigar, 1998) to capture the probability density of electron clouds.

## 6 CONCLUSIONS

In this paper we introduced MCF, where we formulate the problem of molecular conformer generation as learning a distribution over continuous fields on molecular graphs. MCF is formulated specifically as a diffusion generative model over fields, which achieves state-of-the-art performance compared with baselines with strong inductive bias when evaluated on molecular generation across different benchmarks. Notably, MCF achieves these results without explicitly modeling geometric properties of molecules like torsional angles, which makes it simpler to understand and scale. MCF presents an exciting avenue for future research on scaling conformer generation to proteins and other macro molecular structures. We further discuss limitations and future works in Appendix A.1.

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

## A APPENDIX

### A.1 LIMITATIONS AND FUTURE WORK

While MCF shows competitive performance in molecular conformer generation, it does encounter limitations and potential improvements for future explorations. One limitation is that our proposed method is computationally expensive. Extensive computations first stem from the Transformer-based (Vaswani et al., 2017) score network. In MCF, we use a PerceiverIO (Jaegle et al., 2022) as score network, an efficient transformer that allows for sub-quadratic compute. Other efficient transformer architectures and tricks like (Dao et al., 2022) can be used to improve training efficiency. The other factor is computational cost during inference. In MCF, we iterate 1000 timesteps to sample a conformer following DDPM (Ho et al., 2020). Experiments in Appendix A.4 further show that efficient sampling strategies, *i.e.* DDIM (Song et al., 2021), can help significantly increase inference efficiency while maintain high-quality in sampled conformers. Other efficient variants of diffusion models like consistency model (Song et al., 2023) as well as distillation approaches (Berthelot et al., 2023) may be adapted to further decrease the sampling to single step. Also, recent works have demonstrated that diffusion generative model can generate samples following Boltzmann distributions when provided with Boltzmann-distributed training data Arts et al. (2023). Driven by this, our proposed MCF can be adapted to generate molecular conformers that follow Boltzmann distributions when trained with corresponding data. Besides, recent flow matching generative model (Lipman et al., 2022) provides the flexibility of mapping between arbitrary distributions and access to exact log-likelihood estimation. Integrating flow matching framework could help sample molecular conformer from Boltzmann distribution instead of standard Gaussian.

Another limitation could be the fact that MCF does not explicitly model the equivariance of the molecular system. Though experiments in the paper have demonstrated that equivariance may not be necessary to achieve competitive performance on conformer generation. MCF may not perform as well as conformer generation when applied to problems with limited data or related to sequential problems like molecular dynamics (MD) simulations. In future work, we plan to extend MCF to conditional inference. For example, molecular docking can be formulated as conformer generation problem conditioned on proteins (Corso et al., 2022). Also, current framework can be expanded to *de novo* drug designs where no molecule information is provided (Hoogeboom et al., 2022). Besides, scaling up our model to large molecules, like proteins, can be of great interest. MCF by nature provides the flexibility to generate from partially observed sample, which can be suitable for designing proteins with known functional motifs (Watson et al., 2023).

### A.2 IMPLEMENTATION DETAILS

In this section we describe implementation details for all our experiments. We also provide hyper-parameters and settings for the implementation of the score field network $\epsilon_\theta$ and compute used for each experiment in the paper.

#### A.2.1 SCORE FIELD NETWORK IMPLEMENTATION DETAILS

The time-step $t$ is incorporated into the score computation by concatenating a positional embedding representation of $t$ to the context and query sets. The specific PerceiverIO settings used in all quantitatively evaluated experiments are presented in Tab. 4. Practically, the MCF network consists of 6 transformer blocks, each containing 1 cross-attention layer and 2 self-attention layers. Each of these layers has 4 attention heads. An Adam (Kingma & Ba, 2015) optimizer is employed during training with a learning rate of $1e-4$. We use EMA with a decay of $0.9999$. A modified version of the publicly available repository is used for PerceiverIO [1]. Since molecules have different number of atoms, we set the number of context and query sets as the number of atoms during training and inference.

#### A.2.2 ATOMIC FEATURES

We include atomic features alongside the graph Laplacians to model the key descriptions of molecules following previous works Ganea et al. (2021); Jing et al. (2022). Detailed features are listed in Tab. 5.

---

[1] https://huggingface.co/docs/transformers/model_doc/perceiver

| Hyper-parameter | GEOM-QM9 | GEOM-DRUGS | Ablation GEOM-QM9 |
|---|---|---|---|
| #eigenfuncs ($k$) | 28 | 32 | 28 |
| #freq pos. embed $t$ | 64 | 64 | 64 |
| #latents | 512 | 512 | 512 |
| #dim latents | 512 | 512 | 256 |
| #model dim | 768 | 1024 | 768 |
| #blocks | 6 | 6 | 6 |
| #dec blocks | 2 | 2 | 2 |
| #self attends per block | 2 | 2 | 2 |
| #self attention heads | 4 | 4 | 4 |
| #cross attention heads | 4 | 4 | 4 |
| batch size | 64 | 128 | 64 |
| lr | $1e-4$ | $1e-4$ | $1e-4$ |
| epochs | 250 | 300 | 250 |

Table 4: Hyperparameters and settings for MCF on different datasets.

| Name | Description | Range |
|---|---|---|
| atomic | Atom type | one-hot of 35 elements in dataset |
| degree | Number of bonded neighbors | $\{x : 0 \leq x \leq 6, x \in \mathbb{Z}\}$ |
| charge | Formal charge of atom | $\{x : -1 \leq x \leq 1, x \in \mathbb{Z}\}$ |
| valence | Implicit valence of atom | $\{x : 0 \leq x \leq 6, x \in \mathbb{Z}\}$ |
| hybrization | Hybrization type | $\{sp, sp^2, sp^3, sp^3d, sp^3d^2, other\}$ |
| aromatic | Whether on a aromatic ring | $\{True, False\}$ |
| num_rings | number of rings atom is in | $\{x : 0 \leq x \leq 3, x \in \mathbb{Z}\}$ |

Table 5: Atomic features included in MCF.

The atomic features are concatenated with graph Laplacian eigenvectors in both context and query inputs.

### A.2.3 COMPUTE

Each model was trained on an machine with 8 Nvidia A100 GPUs, we trained models for 500 epochs.

### A.2.4 EVALUATION METRICS

Following previous works Xu et al. (2022); Ganea et al. (2021); Jing et al. (2022), we apply Average Minimum RMSD (AMR) and Coverage (COV) to measure the performance of molecular conformer generation. Let $C_g$ denote the sets of generated conformations and $C_r$ denote the one with reference conformations. For AMR and COV, we report both the Recall (R) and Precision (P). Recall evaluates how well the model locates ground-truth conformers within the generated samples, while precision reflects how many generated conformers are of good quality. The expressions of the metrics are given in the following equations:

$$\text{AMR-R}(C_g, C_r) = \frac{1}{|C_r|} \sum_{\mathbf{R} \in C_r} \min_{\hat{\mathbf{R}} \in C_g} \text{RMSD}(\mathbf{R}, \hat{\mathbf{R}}), \tag{3}$$

$$\text{COV-R}(C_g, C_r) = \frac{1}{|C_r|} |\{\mathbf{R} \in C_r | \text{RMSD}(\mathbf{R}, \hat{\mathbf{R}}) < \delta, \hat{\mathbf{R}} \in C_g\}|, \tag{4}$$

$$\text{AMR-P}(C_r, C_g) = \frac{1}{|C_g|} \sum_{\hat{\mathbf{R}} \in C_g} \min_{\mathbf{R} \in C_r} \text{RMSD}(\hat{\mathbf{R}}, \mathbf{R}), \tag{5}$$

$$\text{COV-P}(C_r, C_g) = \frac{1}{|C_g|} |\{\hat{\mathbf{R}} \in C_g | \text{RMSD}(\hat{\mathbf{R}}, \mathbf{R}) < \delta, \mathbf{R} \in C_r\}|, \tag{6}$$

where $\delta$ is a threshold. In general, a lower AMR scores indicate better accuracy and a higher COV score indicates a better diversity for the generative model. Following Jing et al. (2022), $\delta$ is set as $0.5$ for GEOM-QM9 and $0.75$ for GEOM-DRUGS.

## A.3 ADDITIONAL EXPERIMENTS

In this section we include additional experiments ablating architecture choices, as well as prediction the ensemble properties of generated conformers.

### A.3.1 ABLATION EXPERIMENTS

In this section we provide an ablation study over the key design choices of MCF. We run all our ablation experiments on the GEOM-QM9 dataset following the settings in GeoMol (Ganea et al., 2021) and Torsional Diffusion (Jing et al., 2022) and described in Sect. 5.2. In particular we study: (i) how does performance behave as a function of the number of Laplacian eigenvectors used in $\varphi(v)$. (ii) How does the model perform without atom features (*e.g.* how predictable conformers are given only the graph topology, without using atom features). Results in Tab. 6 show that the graph topology $\mathcal{G}$ encodes a surprising amount of information for sampling reasonable conformers in GEOM-QM9, as shown in row 2. In addition, we show how performance of MCF changes as a function of the number of eigen-functions $k$. Interestingly, with as few as $k = 2$ eigen-functions MCF is able to generate consistent accurate conformer.

| | | Recall | | | | Precision | | | |
| | | Coverage ↑ | | AMR ↓ | | Coverage ↑ | | AMR ↓ | |
| $k$ | Atom Features | mean | median | mean | median | mean | median | mean | median |
|---|---|---|---|---|---|---|---|---|---|
| 28 | YES | 94.86 | 100.00 | 0.125 | 0.081 | 91.49 | 100.00 | 0.175 | 0.122 |
| 28 | NO | 90.70 | 100.00 | 0.187 | 0.124 | 79.82 | 93.86 | 0.295 | 0.213 |
| 16 | YES | **94.87** | 100.00 | **0.139** | 0.093 | **87.54** | 100.00 | 0.220 | 0.151 |
| 8 | YES | 94.28 | 100.00 | 0.162 | 0.109 | 84.27 | 100.00 | 0.261 | 0.208 |
| 4 | YES | 94.57 | 100.00 | 0.145 | 0.093 | 86.83 | 100.00 | 0.225 | 0.151 |
| 2 | YES | 93.15 | 100.00 | 0.152 | **0.088** | 86.97 | 100.00 | **0.211** | **0.138** |

Table 6: Experiments on GEOM-QM9 with different numbers of eigenvectors. In these experiments, we use a smaller model than what we have in Table 1

### A.3.2 ARCHITECTURAL CHOICES

To further investigate the design choices of architecture in proposed MCF, we include additional experiments on GEOM-QM9 as shown in Tab. 7. To investigate the effectiveness of using Laplacian eigenvectors as positional embedding, we leverage SignNet (Lim et al., 2022) as the positional embedding, which explicitly models symmetries in eigenvectors. Using SignNet does not benefit the performance when compared with the standard MCF. Though adding edge attributes in SignNet achieves better performance than SignNet alone, the performance is still not rival. Also, it's worth mentioning that SignNet includes graph neural networks (Xu et al., 2018) and Set Transformer (Lee et al., 2019) which makes training less efficient.

In addition, we also report results using a vanilla Transformer encoder-decoder (Vaswani et al., 2017) as the backbone instead of PerceiverIO (Jaegle et al., 2022). The Transformer contains 6 encoder layers and 6 decoder layers with 4 attention heads. Other model hyperparameters follow the same setting as listed in Tab. 4. It is indicated that Transformer is performing worse than PerceiverIO on conformer generation, which validates the design choice of architecture in MCF.

### A.3.3 ENSEMBLE PROPERTY PREDICTION

To fully assess the quality of generated conformers we also compute chemical property resemblance between the synthesized and the authentic ground truth ensembles. We select a random group of 100 molecules from the GEOM-DRUGS and produce a minimum of $2K$ and a maximum of 32 conformers for each molecule following (Jing et al., 2022). Subsequently, we undertake a comparison

| | Precision | | | | Recall | | | |
| --- | --- | --- | --- | --- | --- | --- | --- | --- |
| | Coverage ↑ | | AMR ↓ | | Coverage ↑ | | AMR ↓ | |
| | mean | median | mean | median | mean | median | mean | median |
| MCF SignNet | 94.1 | 100.0 | 0.153 | 0.098 | 87.5 | 100.0 | 0.222 | 0.152 |
| MCF SignNet (edge attr.) | 95.3 | 100.0 | 0.143 | 0.091 | 90.2 | 100.0 | 0.197 | 0.135 |
| MCF Transformer | 94.3 | 100.0 | 0.159 | 0.111 | 90.7 | 100.0 | 0.202 | 0.136 |
| MCF PerceiverIO | **96.57** | **100.00** | **0.107** | **0.072** | **94.60** | **100.00** | **0.130** | **0.084** |

Table 7: Molecule conformer generation results with different network architectures on GEOM-QM9.

| | $E$ | $\mu$ | $\Delta\epsilon$ | $E_{\min}$ |
| --- | --- | --- | --- | --- |
| OMEGA | 0.68 | 0.66 | 0.68 | 0.69 |
| GeoDiff | 0.31 | 0.35 | 0.89 | 0.39 |
| GeoMol | 0.42 | 0.34 | 0.59 | 0.40 |
| Torsional Diff. | **0.22** | 0.35 | **0.54** | 0.13 |
| MCF | 0.66±0.05 | **0.26±0.05** | 0.64±0.06 | **0.02±0.00** |
| Torsional Diff. (our eval) | 3.07±2.32 | 0.61±0.38 | 1.71±1.69 | 4.11±7.91 |
| MCF (ours) | 1.00±0.70 | 0.44±0.36 | 1.32±1.40 | 1.16±2.02 |

Table 8: Median averaged errors of ensemble properties between sampled and generated conformers ($E$, $\Delta\epsilon$, $E_{\min}$ in kcal/mol, and $\mu$ in debye).

of the Boltzmann-weighted attributes of the created and the true ensembles. To elaborate, we calculate the following characteristics using xTB (as documented by (Bannwarth et al., 2019)): energy ($E$), dipole moment ($\mu$), the gap between HOMO and LUMO ($\Delta\epsilon$), and the lowest possible energy, denoted as $E_{\min}$. Since we don't have the access to the exact subset of DRUGS used in Jing et al. (2022), we randomly pick three subsets and report the averaged and standard deviation over three individual runs with different random seeds. The results are listed in Tab. 8. Our model achieves the lowest error on $E_{\min}$ when compared with other baselines, which demonstrates that MCF is succeeds at generating stable conformers that are very close to the ground states. This could root from the fact that MCF doesn't rely on rule-based cheminfomatics methods and the model learns to better model stable conformers from data. Besides, MCF achieves competitive performance on $\mu$ and $\Delta\epsilon$. However, the error of $E$ is high compared to the rest of approaches, meaning that though MCF performs well in generating samples close to ground states, it may also generate conformers with high energy that are not plausible in the dataset.

To further evaluate the performance on ensemble properties, we randomly pick 10 molecules from test set of GEOM-DRUGS and compare MCF with our replica Torsional Diffusion on the subset as shown in the last two rows of Tab. 8. We use the checkpoints from the public GitHub repository[2] of Torsional Diffusion to sample conformers. Unlike previous setting which only sample 32 conformers, we sample $2K$ conformers for a molecule with $K$ ground truth conformers. We report the average and standard deviation of errors over the 10 molecules. It is indicated that MCF generates samples with ensemble properties that are closer to the ground truth. Fig. 7 shows the conformers with lowest and highest energy in ground truth, MCF samples, and Torsional Diffusion samples. MCF generates samples conformers with ground states close to the ground truth. Sometimes it can even find conformers with lower energy (e.g. Fig. 7(c)). However, both MCF and Torsional Diffusion can generate conformers with higher energies than ground truth.

## A.4 EFFICIENT SAMPLING

To reduce the computational cost of MCF in inference, we apply efficient sampling technique, i.e. DDIM (Song et al., 2021), with significantly smaller number of sampling steps. Tab. 9 show the comparison of MCF and Torsional Diffusion (Jing et al., 2022) with different sampling steps. With 20 or 50 sampling steps, MCF achieves comparable performance as the DDPM with 1000 sampling steps and greatly surpasses Torsional Diffusion with same sampling steps. Indeed, with very small sampling

---

[2]https://github.com/gcorso/torsional-diffusion

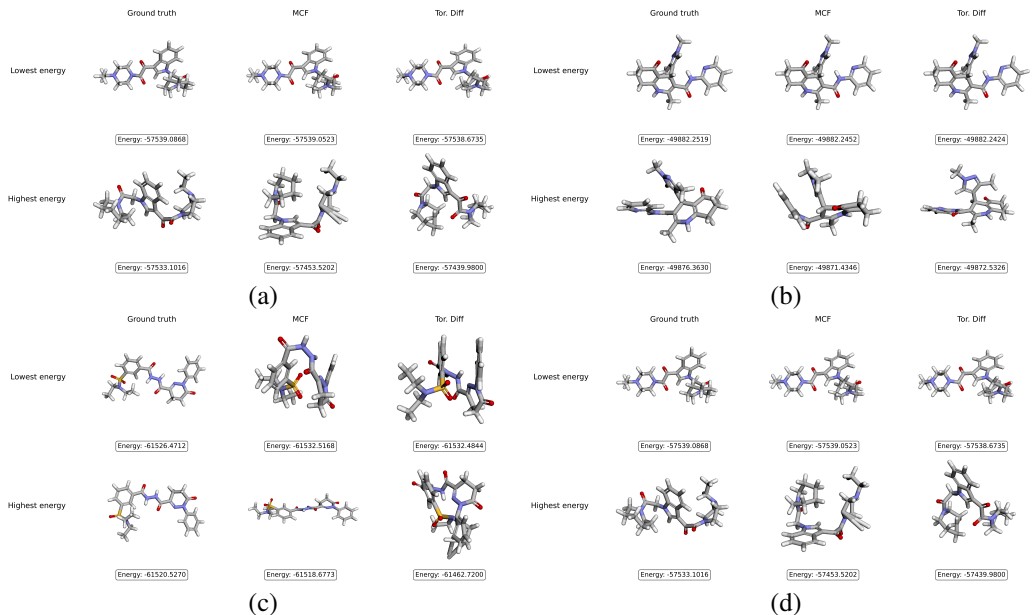

Figure 7: Examples of conformers with lowest and highest energies in ground truth, MCF samples, and Torsional Diffusion samples for different molecules.

| | steps | Precision | | | | Recall | | | |
|---|---|---|---|---|---|---|---|---|---|
| | | Coverage ↑ | | AMR ↓ | | Coverage ↑ | | AMR ↓ | |
| | steps | mean | median | mean | median | mean | median | mean | median |
| Torsional Diff. | 3 | 42.9 | 33.8 | 0.820 | 0.821 | 24.1 | 11.1 | 1.116 | 1.100 |
| Torsional Diff. | 5 | 58.9 | 63.6 | 0.698 | 0.685 | 35.8 | 26.6 | 0.979 | 0.963 |
| Torsional Diff. | 10 | 70.6 | 78.8 | 0.600 | 0.580 | 50.2 | 48.3 | 0.827 | 0.791 |
| Torsional Diff. | 20 | 72.7 | 80.0 | 0.582 | 0.565 | 55.2 | 56.9 | 0.778 | 0.729 |
| Torsional Diff. | 50 | 73.1 | 80.4 | 0.578 | 0.557 | 57.6 | 60.7 | 0.753 | 0.699 |
| MCF$_{\text{DDIM}}$ | 3 | 15.05 | 0.00 | 1.032 | 1.041 | 5.71 | 0.00 | 1.521 | 1.525 |
| MCF$_{\text{DDIM}}$ | 5 | 42.86 | 35.50 | 0.813 | 0.814 | 20.07 | 11.54 | 1.149 | 1.149 |
| MCF$_{\text{DDIM}}$ | 10 | 74.14 | 83.25 | 0.610 | 0.601 | 48.95 | 46.35 | 0.841 | 0.813 |
| MCF$_{\text{DDIM}}$ | 20 | 80.87 | 88.89 | 0.522 | 0.504 | 59.72 | 60.23 | 0.745 | 0.708 |
| MCF$_{\text{DDIM}}$ | 50 | 81.87 | 88.89 | 0.481 | 0.459 | 62.01 | 62.53 | 0.708 | 0.660 |
| MCF$_{\text{DDIM}}$ | 100 | 81.97 | 89.10 | 0.466 | 0.441 | 62.81 | 63.64 | 0.693 | 0.641 |
| MCF$_{\text{DDPM}}$ | 1000 | 81.62 | 89.22 | 0.468 | 0.438 | 61.63 | 62.50 | 0.705 | 0.650 |

Table 9: Molecule conformer generation results with different sampling strategies and timesteps on GEOM-DRUGS.

steps (*e.g.* 3 or 5 steps), Torsional Diffusion demonstrate better performance than MCF. We attribute this to the fact that Torsional Diffusion uses a pre-processing step in which a cheminformatics method predicts local substructure in 3D spaces before applying their model. So even before inference the cheimformatics prediction provides a strong prior. It should also be noted that with small sample steps, Torsional Diffusion also suffer from low sample quality compared with the counterparts with more steps. Fig. 8 show some examples of sampled conformers from MCF with different sampling steps. Even with small sampling steps, MCF can still generate plausible 3D structures, especially for the heavy atoms.

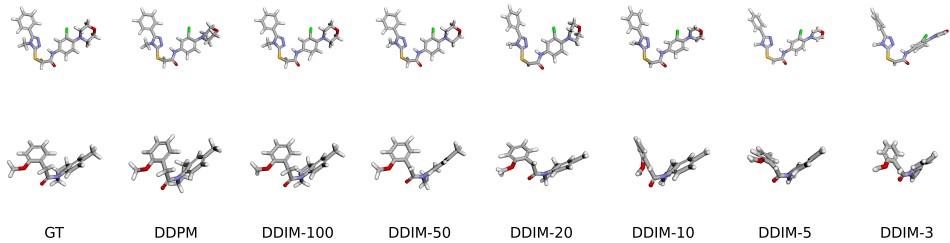

GT    DDPM    DDIM-100    DDIM-50    DDIM-20    DDIM-10    DDIM-5    DDIM-3

Figure 8: Examples of conformers with different sampling steps.

## B    ADDITIONAL VISUALIZATION

In the supplementary material we provide videos showing the iterative inference process of MCF on different molecules in GEOM-QM9, GEOM-DRUGS and GEOM-XL. Finally in Fig. 9 we show GT and generated conformers from MCF for a molecule in GEOM-DRUGS.

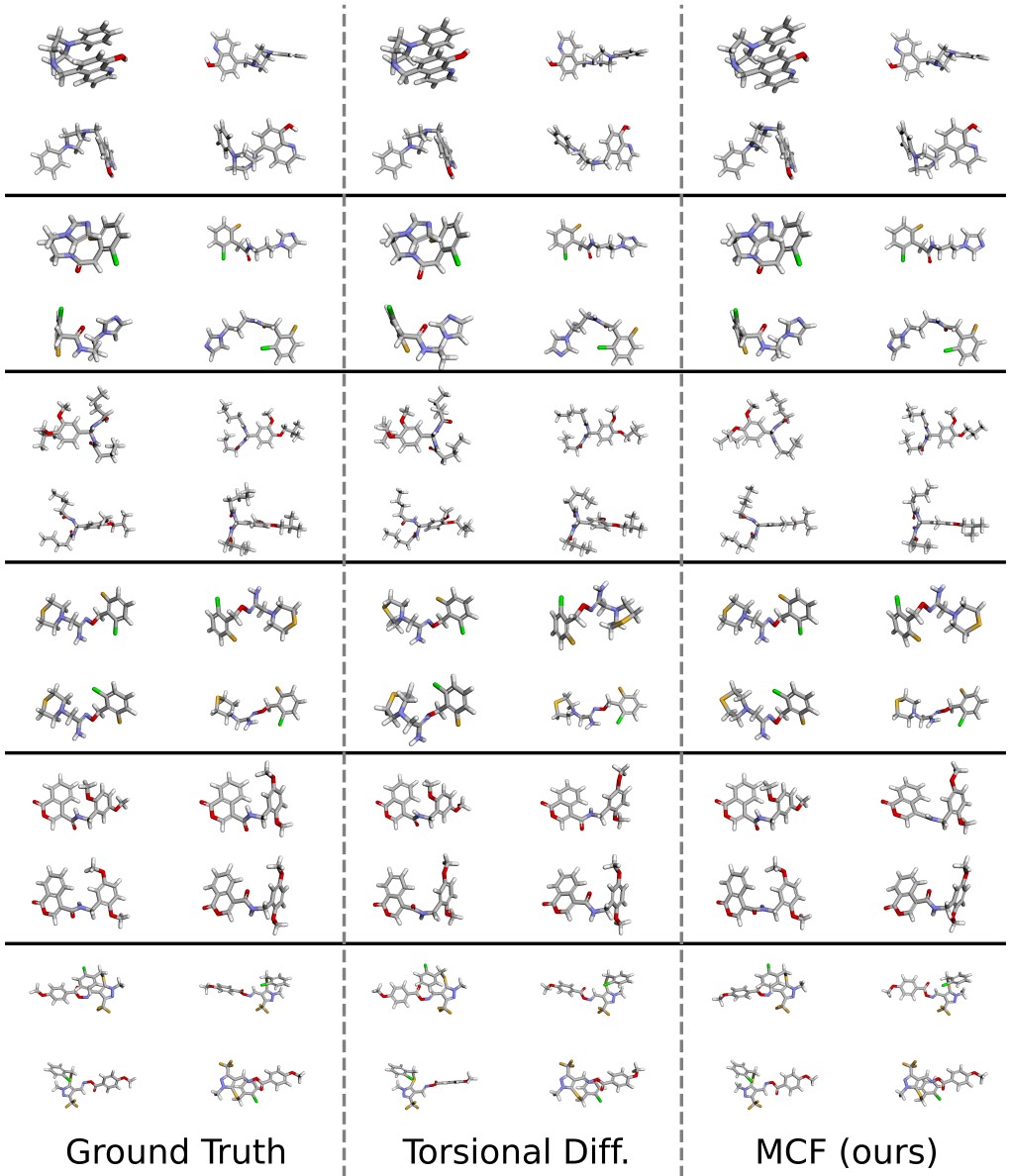

Figure 9: GT and generated conformers from MCF for a molecule in GEOM-DRUGS.

