# OpenReview forum: "Generating Molecular Conformer Fields"
_ICLR.cc/2024/Conference — Submitted to ICLR 2024_

### Official Review · Reviewer_6EV9 · 2023-10-30

**Soundness:** 2 fair
**Presentation:** 3 good
**Contribution:** 3 good
**Rating:** 5
**Confidence:** 4

**Summary:**

The authors apply Denoising Probabilistic Fields (DPFs) to the problem of molecular-conformer generation. This entails training a diffusion model where molecular node identities (given as Lapalcian eigenvectors) are given to the diffusion model in order to denoise associated 3D coordinates of each node. The authors show competitive results with another diffusion-based method (Torsional Diffusion) on three standard molecular-conformer datasets.

**Strengths:**

### Shows comparable performance to Torsional Diffusion (state of the art)

The authors have shown using three standard datasets: GEOM-QM9, GEOM-DRUGS, and GEOM-XL, that their method of DPFs on molecular conformers is comparable. The performance metrics used encompass recall and precision and matches that used in previous work. The comparisons seem solid for the most part. Overall, although these results do not consistently outperform the state of the art, they look comparable depending on the dataset.

### Demonstrates that equivariant architectures are not strictly necessary for this problem

The authors also call attention to the fact that DPFs get comparable performance to Torsional Diffusion, and this shows that roto-translational equivariance may not be strictly required for conformer prediction (Torsional Diffusion is implemented using an SE(3) equivariant architecture, but the DPF in this manuscript directly generates 3D coordinates of atoms from molecular graphs. Although this is not a core contribution of the paper, it is useful to know nonetheless.

Also, I very much enjoyed and appreciated the quotations in “‘simple’ DDPM loss”.

**Weaknesses:**

### Limited novelty relative to DPF paper

The technical and empirical novelty is fairly limited in this paper. It is a direct application of DPFs (Zhuang et. al. 2023) to a molecular-conformer generation. The neural network also has an identical architecture, and the difference is solely to apply it to this different problem. The representation of atom/node “coordinates” (i.e. the field domain) as Laplacian eigenvectors is also a direct application of Maskey, et. al. 2022.

### Several claims are not very well substantiated; these claims should be reworded in the text

Below are a list of claims I found in the paper which I did not find to be well substantiated. The authors should either provide stronger evidence for these claims, or reword the claim to be more accurate.

-  The paper claims that explicitly enforcing domain-specific inductive biases (e.g. periodic domain of possible torsional angles in Torsion Diff) comes at a cost, but what is the cost? Torsional Diffusion seems to be performing just as well as MCF (sometimes better).

- The paper claims that MCF is more scalable than other methods (e.g. Torsional Diffusion). The architecture of MCF is quite large, however, and for a system of 100 atoms, there are 200 queries to the model (100 context pairs and 100 query pairs). Is it really more scalable than Torsional Diffusion?

- The paper claims that the ablation study shows what factors are important for molecular-conformer generation, but this is a pretty grandiose claim. The ablation study only shows the impact of atom features and the size of the Laplacian eigenvectors, _specifically on MCF performance_. It does not lend insight into what factors are important for molecular-conformer generation in general.

- The paper claims that explicitly enforcing roto-translation equivariance is not a strong requirement for generalization, and that equivariance is not particularly useful for molecular-conformer generation. Although the authors have shown similar and comparable performance between an equivariant and non-equivariant architecture, it is not yet clear whether or not equivariance really affects performance. For example, one could ask whether MCFs could outperform Torsional Diffusion if it had used a roto-translationally equivariant architecture (not just PerceiverIO).

### Splitting of molecules into datasets may have leakage

The datasets seem to be split into training/validation/test uniformly at random, but this can lead to train/test leakage. Many molecules in the datasets share very similar scaffolds, and it is well known that prediction between similar scaffolds is much easier. To really assess the quality of the model, the datasets should be split into scaffold-aware subsets to avoid leakage.

I understand this issue is almost certainly inherited from previous works (e.g. Torsional Diffusion), so I do not count this against the evaluation of this paper, but I highly recommend exploring better dataset splits so that we may all do better moving forward.

### The continuity of MCF needs more exploration and explanation

Figure 6 and the exploration of the continuity of MCFs does not seem very strong or insightful. The analysis only shows that the model learns to “round” coordinates toward atom centers. There is no indication that the model actually learns any information about molecular bonds (and not to put atom density on them). One could claim that the model will always output some set of atomic coordinates which are not too close to each other _regardless_ of the input, even if the network has no understanding of bonds.

It is also not clear what the benefit is to being able to predict on interpolated atom coordinates. Is the goal to be able to input the interpolated eigenvector halfway between two nodes and generate the 3D coordinate that is halfway between the two atoms? If so, then this analysis very much shows the opposite. Is the goal to be able to input the interpolated eigenvector halfway between two nodes and probabilistically generate the 3D atomic coordinate of either endpoint atom? If so, then this analysis does somewhat show that, but I don’t see why this latter question is interesting or useful.

### Some parts of the writing can be clearer

- More background on DPF would be nice; this work is so dependent on DPFs, that it would be good to have more background on how they work

- Limitations of this method should really go in the main text

- It would be good to include the definitions of performance metrics like RMSD recall (in the supplement) instead of needing to refer to the Torsional Diffusion paper

- Notation should be cleaner: if a molecule’s conformer is a function $f: \mathcal{G}\rightarrow\mathbb{R}^{3}$, this implies the function maps an entire molecular graph to a single 3D coordinate

- “equivairant” in first paragraph of Section 2

- “In contraposition” should be “In contrast”

**Questions:**

### Is the dataset split the same as in previous works? Is the RMSD cutoff $\delta$ the same as previous works for precision/recall?

It is crucial that the dataset splits (and RMSD precision/recall $\delta$ value) are the same here compared to prior works, because the performance numbers in the tables are directly copied from those papers.

### What atom features are concatenated? How are bond features incorporated (e.g. double vs triple bonds)?

### What is the purpose of predicting on the query set instead of context set in DPFs?

This question is more related to DPFs in general (and is not specific to MCFs), to make sure I understand the work. Why is there a separate query set and context set in DPFs? If we are adding noise to the signal of the context set, why not just predict the denoising of the context? Instead, we are taking another query set (with the same coordinates) and adding noise separately and predicting denoising of the query set.

---

> ### Author Response · Authors · 2023-11-15
> **Official Comments by Authors**
>
> We thank the reviewers for the constructive reviews. Please find below our detailed responses to the weaknesses and questions:
>
> 1. **Limited novelty relative to DPF paper**
>    - In this work we generalize the problem setting in DPF to deal with functions on graphs (as opposed to functions on Euclidean spaces). Our proposed approach is conceptually simple to implement yet very powerful in its application. **Note that MCF sets the current state-of-the-art for conformer generation.** We believe that approaches that are both simple to implement and powerful in their empirical results tend to have the longest lasting impact in the field.
>
> 2. **Several claims are not very well substantiated**
>
>    - **What is the cost of enforcing domain-specific inductive biases (e.g. periodic domain of possible torsional angles in Torsion Diff)**
>       * Torsional diffusion requires breaking molecules into fragments connected by rotatable bonds. This process relies on rule-based methods that can fail, especially for complex molecules. For example, in our replica of torsional diffusion on GEOM-XL using their public codebase, we only successfully generate conformers of 77 molecules out of 102 in the database. Besides, torsional diffusion utilizes cheminformatic methods to predict the local 3D structures. Therefore, the accuracy of the model is dependent on the local models and may neglect effects of long-range interatomic interactions from other substructures. We have further clarified the cost in Introduction section.
>       * Furthermore, we trained an MCF model with updated hyper-parameters as reflected in Tab. 4. We applied updated latent dimensions as well as number of latents and trained the model for more epochs. This model now obtains state-of-the-art performance,  outperforming torsional diffusion on challenging the GEOM-DRUGS and GEOM-XL datasets (shown in updated Tab. 2 & 3).
>
>    - **Is it really more scalable than Torsional Diffusion?**
>       * First, more scalable means MCF operates directly on raw data in 3D Euclidean space without the domain-specific efforts to pre-processing the data. Such pre-processing can lead to failure cases for complex molecules and forces the conformer generation adhered to other substructure models. These priors can prevent the models from scaling up to larger and more complex molecular systems. In addition, MCF is built upon PerceiverIO, an attention-based Transformer model. PerceiverIO includes a latent array that cross attends with input context and output query. In our experiments, we set the number of latent as 512. For a molecule with 100 atoms, the model has degree of freedom up to 512 which is much greater the number of atoms. Both aspects provide the flexibility of our proposed MCF to scale up.
>
>    - **The paper claims that the ablation study shows what factors are important for molecular-conformer generation, but this is a pretty grandiose claim.**
>       * We thank the reviewer for pointing this out, this is a misscommunication that is clarified in the updated version of the paper. Our claim is to show what implementation factors and node features are important for MCF to obtain a good performance. We have rephrased the claim to clarify our contributions.
>
>    - **The paper claims that explicitly enforcing roto-translation equivariance is not a strong requirement for generalization, and that equivariance is not particularly useful for molecular-conformer generation. Although the authors have shown similar and comparable performance between an equivariant and non-equivariant architecture, it is not yet clear whether or not equivariance really affects performance. For example, one could ask whether MCFs could outperform Torsional Diffusion if it had used a roto-translationally equivariant architecture (not just PerceiverIO).**
>       * We have updated Tab. 2 after training an MCF model with updated hyper-parameters (see Tab. 4). MCF now surpasses all baseline models by a large margin on GEOM-DRUGS. The latest results further demonstrate that roto-translational equivairance is not a strong requirement for achieving superior performance on molecular conformer generation. Our model, which is built on a general-purposed Transformer-based architecture, can achieve better performance than other models which are roto-translational equivariant.
>       * Also, we would like to underline that to the best of our knowledge, our work is the first to highlight the question whether equivariance is required for solving molecular conformer generation problem. Previous works have focused on building equiavriant neural networks and baking inductive biases (e.g. modeling torsional angles) for conformation generation. Indeed current work doesn’t completely answer the question whether equivariance is needed for conformer generation. But we hope this work can shed a light on future investigations of the role equivariance plays in such tasks.
>
> ------- see following comments below -------

---

> > ### Author Response · Authors · 2023-11-15
> > **Official Comments by Authors (2)**
> >
> > ------- continued comments -------
> >
> > 3. **The continuity of MCF needs more exploration and explanation**
> >    - We thank the reviewer for raising this point. Our goal is to show that one can evaluate MCF with an interpolated eigen-function coordinate. Our observation is that if one has supervision for what are corresponding 3D points for interpolated eigen-functions (eg. simulating the PDF of electron clouds between atoms) those could be used during during training. In addition, we see that MCF does not completely snap generated 3D coordinates to atom positions and there are a few 3D points generated in between atoms (colored in light blue in Fig 6.). This is surprising because MCF was never trained to produce any logical output for interpolated eigen-functions.
> >
> > 4. **Some parts of the writing can be clearer**
> >
> >    - **More background on DPF would be nice; this work is so dependent on DPFs, that it would be good to have more background on how they work**
> >       * We thank the reviewer for the suggestion and have added the DPF background to section 3.1 of Preliminaries in the revised main text.
> >
> >    - **Limitations of this method should really go in the main text**
> >       * We thank the suggestion. However, adding limitations to the main text leads to exceeding the maximum number of pages. We have added a note in the Conclusion section that points the readers to the Limitations and Future Works section in Appendix.
> >
> >    - **It would be good to include the definitions of performance metrics like RMSD recall (in the supplement) instead of needing to refer to the Torsional Diffusion paper**
> >       * We thank the reviewer for bring this up. We have added the definitions of the metrics to Appendix A.2.4.
> >
> >    - **Notation should be cleaner: if a molecule’s conformer is a function $f: \mathcal{G} \rightarrow \mathbb{R}^3$, this implies the function maps an entire molecular graph to a single 3D coordinate**
> >       * Here $f: \mathcal{G} \rightarrow \mathbb{R}^3$ represents an implicit function/field that maps  points on the graph $\mathcal{G}$ to  Euclidean space $\mathbb{R}^3$. In this case, $f$ doesn’t denote the mapping from a whole molecular graph to its 3D conformer but rather a node in the molecular graph. We have added extra explanations in the updated manuscript to clarify the concept.
> >
> >    - **“equivairant” in first paragraph of Section 2. “In contraposition” should be “In contrast”**
> >       * We thank the reviewer for pointing the typos and we have fixed them in the updated manuscript.
> >
> > 5. **Is the dataset split the same as in previous works? Is the RMSD cutoff $\delta$ the same as previous works for precision/recall?**
> >    - Yes, we use the exact same data splitting as reported in GeoMol and Torsional Diffusion. And following Torsional Diffusion, we use 0.5 and 0.75 as coverage cutoff for GEOM-QM9 and GEOM-DRUGS respectively. We have added the clarification to Appendix A.2.4.
> >
> > 6. **What atom features are concatenated? How are bond features incorporated (e.g. double vs triple bonds)?**
> >    - We include atomic number, aromatic, degree, hybridization, number of rings, formal charge, and implicit valence in the atom feature following [1]. We have added an extra section A.2.2 in Appendix to clarify the atomic features used in this work. In MCF, we don’t explicitly include bond features since the bond information is encoded by the atom features (e.g., degree, aromatic, formal charge, etc) and the topological information from graph Laplacians. As we shown in Appendix A.2.2 and Tab. 5, adding extra atom or bond features doesn’t lead to significant performance improvement.
> >
> > 7. **What is the purpose of predicting on the query set instead of context set in DPFs?**
> >    - Prediction on query set instead of context set follows the idea of building generative model on fields. As a continuous field, one would expect it capable of mapping any points in the input space to the corresponding output space. To this end, the score function is built to predict the signal of any point in the input space (query set) given some observations in the input space (context set). The context set contains pairs of input and output that provides the information of the fields while the query set contains points that one expect to evaluate on. Such a framework provides the flexible of learning the distribution of fields over the continuous input space.
> >
> > ------- see following comments below -------

---

> > > ### Author Response · Authors · 2023-11-15
> > > **Official Comments by Authors (3)**
> > >
> > > ------- continued comments -------
> > >
> > > 8. **Splitting of molecules into datasets may have leakage**
> > >    - We agree with the review that random splitting the molecular data can lead to “leakage” in a way, namely molecules with similar scaffolds can be found in both training and test sets. In this work, we use the exact same random splitting  as in previous works (e.g., Torsional Diffusion, GeoMol, etc.) so that we can have an apples-to-apples comparison with the baseline models.
> > >    - Besides experiments on standard random splitting of GEOM-QM9 and GEOM-DRUGS datasets. We also benchmark on GEOM-XL (see Tab. 3), a dataset that contains molecules with number of atoms greater than 100 collected by [1]. MCF trained on GEOM-DRUGS achieves superior performance on GEOM-XL than the baselines. Notice the median number of atoms in GEOM-DRUGS is 40. The results demonstrate that our proposed MCF can generalize to molecules with different scaffolds.
> > >
> > > References:
> > >
> > > [1] Jing, B., Corso, G., Chang, J., Barzilay, R. and Jaakkola, T., 2022. Torsional diffusion for molecular conformer generation. Advances in Neural Information Processing Systems, 35, pp.24240-24253.

---

> > > > ### Comment · Reviewer_6EV9 · 2023-11-22
> > > >
> > > > Thank you to the authors for providing additional clarification and results. I also very appreciate the highlighting of changes to the manuscript in red.
> > > >
> > > > Several of the initial concerns have been alleviated, and so here are some of the remaining concerns.
> > > >
> > > > **The cost of enforcing domain-specific inductive biases**
> > > >
> > > > My understanding of this claim is that methods like TorsionDiff can fail for larger molecules, or other cheminformatic methods may be too limited to whatever these cheminformatic biases are. Is there evidence that MCF does better in specifically in these areas? Especially because the technical novelty of MCF is somewhat limited, it is important to understand how MCF improves on these other methods in different areas (and where it is less appropriate).
> > > >
> > > > **Scalability compared to TorsionDiff**
> > > >
> > > > Can the authors provide some understanding on the amount of time taken (in number of training epochs, training time, and sampling time) between MCF and TorsionDiff? One concern is that DPF effectively relies on twice as much diffusion work due to the context set and query set.
> > > >
> > > > **Continuity of MCF**
> > > >
> > > > Of course, one could take a trained MCF model and input any eigenfunction whatsoever, but the output may not be particularly meaningful. MCF does seem to *tend* to snap 3D coordinates to atomic positions, and less between atoms. One could argue that this is just the effect of the model only ever seeing atomic positions as the output, and so the model only ever outputs these positions, no matter what. Any predictions that lie between atoms may just be the result of OOD instability. And it is precisely because MCF was never trained to produce any logical output for interpolated eigenfunctions that it is very difficult to believe it has learned anything meaningful about atomic coordinates other than memorizing that it should always generate atomic positions and never in between. That is to say, the only thing that can be reasonably concluded from this analysis is, "MCF was trained to generate atomic coordinates, and it tends to generate atomic coordinates; even if the input is an interpolated between graph nodes, MCF will still tend to round to atomic coordinates". This conclusion speaks to some amount of stability for the model, but I don't see how it is particularly interesting or useful. In this light, this behavior might even be considered a *limitation* of the method, because it is reasonable to expect that inputting an interpolated eigenfunction between nodes should generate the corresponding coordinate between atoms.

---

> ### Author Response · Authors · 2023-11-22
> **Official Comment by Authors**
>
> We thank the reviewer for engaging with the discussion and bringing up valuable comments. Please find the our detailed replies below:
>
> 1. The cost of enforcing domain-specific inductive biases
>
> - Torsional diffusion relies on cheminformatic methods to find the rotatable bonds and predict the 3D structure of substructures. As the reviewer mentioned, such cheminformatic methods can fail in some cases, especially for large molecules. In particular, in GEOM-XL dataset, which contains large molecules with more than 100 atoms, Torsional Diffusion only successfully generates 77 out of 102 molecules when reproduced using their public codebase. While MCF can generate conformers of all molecules without failure cases. This provides an example of the cost of domain-specific inductive biases and the advantages of MCF with simple architecture and training recipe. MCF achieves better performance than Torsional Diffusion on GEOM-XL while successfully generating conformers for all molecules. To the best of our knowledge, our work is the first to raise the question of whether strong inductive bias is required in achieving superior performance on molecular conformer generation.
>
> 2. Scalability compared to TorsionalDiff
>
> - First, as discussed in the previous question, MCF can generate conformers for large molecules where Torsional Diffusion. This indicates the capability of scaling MCF to large molecules. Second, in terms of computational cost, we have added the experiments of using efficient sampling technique (i.e. DDIM) in Appendix of updated manuscript. It shows that using the same 20 denoising steps (which is the same number of steps as main results reported in Torsional Diffusion), MCF achieves significantly better performance than Torsional Diffusion. Under this setting, MCF takes approximately 0.5s on one A100 GPU and 100s on a CPU to sample one conformer in GEOM-DRUGS. Given the common accessibility of GPUs, we believe benchmark on GPU time is a reasonable practice, which is much faster than acquiring the ground truth conformers using DFT. Also, it should be noted that the benchmark was conducted with batch size 1. In practice, one can always stack multiple molecules as a batch and further improve the sampling efficiency. In terms of training cost, MCF is trained for 300 epochs on GEOM-DRUGS on 8 A100 GPUs. Indeed, it is more costly than Torsional Diffusion in training. But it should be noted that training is a one-time cost and **MCF sets the new state-of-the-art on standard molecular conformer generation tasks**. Lastly, since MCF doesn’t rely heavily on cheminformatic methods to generate conformers, we hope the framework can be scaled to large molecules like proteins in the future work.
>
> 3. Continuity of MCF
>
> - We agree with the reviewer that current results don’t reflect physically meaningful structures and it’s still an open question to be investigated. Here, we include the experiment of interpolated eigenfunctions to conceptually investigate the flexibility of defining conformer generation problem as a field. It's an interesting observation that MCF can generate feasible conformers even when the input interpolated eigenfunctions have never been seen during training. Such that MCF is not over-fitted to certain eigenfunctions. It provides a hint that MCF may learn to generate distributional aspects of atomic positions purely from correlations from training data. Admittedly, we recognize this is highly speculative and needs further empirical investigation to substantiate in future works. Also, when provided molecular conformer data with distribution of electron density, MCF may be extended to predict electron density beyond atomic positions in future works as well. Following the suggestion, we will move the discussion to Appendix and further clarify the scope of the experiments.

---

### Official Review · Reviewer_ysLZ · 2023-11-01

**Soundness:** 3 good
**Presentation:** 2 fair
**Contribution:** 2 fair
**Rating:** 6
**Confidence:** 3

**Summary:**

This paper proposes a new approach Molecular Conformer Fields (MCF) for molecular conformation generation.  The approach is based on   Diffusion Probabilistic Fields (Zhuang et al., 2023), which learns a distribution of functions (fields) with DDPM. The main difference is that the authors compute normalized graph Laplacian to adopt DPF on the graph-structure molecular data. The experiments on GEOM-QM9 and GEOM-Drugs show that the proposed method is very effective, which can outperform or on-par with existing SOTA models, even though the proposed method directly predicts 3D atom positions instead of torsion angles or other domain-specific variables.

**Strengths:**

It's exciting to see that the proposed method can achieve better performance than TorsionDiff by directly predicting 3D atom positions on the GEOMS benchmark. The empirical results demonstrate the effectiveness of the proposed method.

**Weaknesses:**

- The proposed method looked novel to me at first, but after I read the DPF paper (Zhuang et al., 2023), I found the novelty is actually limited. The authors compute normalized graph Laplacian as the "index" of atom in the graph,  an analogy of the pixel (x, y) index of the image. The main formulation and model architecture are same to DPF.
- In terms of writing, I think more high-level introduction to DPF is needed in the background / preliminary section. The construction of context pairs and query pairs appear at a sudden, without any motivation behind them, which makes the paper hard to read.

**Questions:**

- Why should the context pair also be corrupted? Is there any motivation behind it?
- In Sec. 3.2 the notation looks problematic. How can a function be used as the variable of Gaussian distribution?
- I don't understand why the proposed method can outperform GeoDiff, both of which are based on DDPM and aim to solve the same task. Does the improvement come from the DPF formulation, the more powerful PerceiverIP architecture, or something else?
- One related work [1] on conformation generation is missing

[1] Guan, J., Qian, W. W., Ma, W. Y., Ma, J., & Peng, J. Energy-inspired molecular conformation optimization. ICLR 2022.

---

> ### Author Response · Authors · 2023-11-15
> **Official Comments by Authors**
>
> We thank the reviewers for the constructive reviews. Please find below our detailed responses to their concerns:
>
> 1. **The main formulation and model architecture are same to DPF**
>    - In this work we generalize the problem setting in DPF to deal with functions on graphs (as opposed to functions on Euclidean spaces). Our proposed approach is conceptually simple to implement while being very powerful in its application. **Note that MCF sets the current state-of-the-art for conformer generation.** We believe that approaches that are both simple to implement and powerful in their empirical results tend to have the longest lasting impact in the field.
>
> 2. **More explanation about background of DPF**
>    - We have included a sub-section 3.1 in the Preliminaries introducing diffusion models and diffusion probabilistic fields.
>
> 3. **Why should the context pair also be corrupted? Is there any motivation behind it?**
>    - During training, the context pair provides the information about the field/function at a particular timestep $t$. Drawing an analogy to the image case, the input to the model at timestep $t$ is a noisy image. Similarly, the input to MCF at timestep $t$ is a set of noisy context pairs that explicitly encode the field. At inference time, sampling from MCF starts from gaussian samples for both context and query pairs. Therefore, noisy pairs are required to effectively model the distribution of fields/functions.
>
> 4. **How can a function be used as the variable of Gaussian distribution?**
>    - In our case, we sample a conformer field $f_0 \sim q(f_0)$. Here, $q(f_0)$ denotes the ground truth distribution of conformer fields, namely the implicit functions that map points from molecular graph space to 3D Euclidean space, which is not a Gaussian distribution. We sample noise from Gaussian distribution $\epsilon \sim N(0, I)$ to corrupt the signal (i.e., 3D atomic coordinates) in context and query pairs rather than the field itself.
>
> 5. **Why can the proposed method outperform GeoDiff?**
>    - As the reviewer mentioned, the superior performance of MCF comparing to GeoDiff roots from both the field formulation and the relaxation of using equivariant constraints in the architecture. MCF models the distribution of fields instead of distributions over discrete graphs. In addition, our backbone architecture (ie. PerceiverIO), is an efficient Transformer version, which can model the global interactions between atoms. On the other hand, GeoDiff is built upon an equivariant graph neural network which is known to suffer from over-smoothing and hard to model long-range interactions. Besides, our transformer backbone provides more flexibility of scaling up the number of parameters. In our case, we implement the number of latents as 512 which widens the bandwidth and degree of freedom in modeling the conformer field beyond number of atoms (which is usually in the order of 10).
>
> 6. **Missing related works**
>    - We thank the reviewer for point us to the related work, we have included that in the revised manuscript.

---

> > ### Comment · Reviewer_ysLZ · 2023-11-16
> > **Problematic Notation**
> >
> > For question 4, of course I understand $q(f_0)$ is the distribution of the conformer fields and not the Gaussian distribution.  My question is: the expression such as $q(f_t | f_{t-1}) = N(f_{t-1} | \sqrt{\bar{\alpha}_t} f_0 + (1 - \alpha_t) I)$ is problematic. As you have said, the noise is added to the signal in the context-query pairs, but such a notation is very misleading.

---

> > > ### Author Response · Authors · 2023-11-16
> > >
> > > We thank the reviewer for clarifying their concern. We borrow this notation from DPF and adopt it for 2 main reasons: a) to minimize introducing new notation that makes it more difficult to understand the differences between DPF and MCF. b) to straightforwardly replace the standard DDPM notation that uses samples $x \in \mathbb{R}^n$ (eg. images) with functions $f: \mathcal{G} \rightarrow \mathbb{R}^3$.  The notation $q(f_t | f_{t-1}) = N(f_{t-1} | \sqrt{\bar{\alpha}_t} f_0 + (1 - \alpha_t) I)$ represents the conditional distribution on the output space (which follows a Gaussian) since the domain of the function $\mathcal{G}$ never changes, only the values assigned to it. Would the reviewer be satisfied if we clarify this in text on the paper?

---

> > > > ### Comment · Reviewer_ysLZ · 2023-11-16
> > > >
> > > > Thank the authors for the quick response. It would be great (and I believe necessary) if you could clarify this in the text of the paper.

---

> > > > > ### Author Response · Authors · 2023-11-17
> > > > > **Official Comment by Authors**
> > > > >
> > > > > We thank the reviewer for the suggestions. We have updated the paper with a clarification on the notation both in Sect 3.1 and 3.2. We kindly remind the reviewer that we have updated results for MCF which is now obtaining state-of-the-art performance across datasets and outperforming Torsional Diffusion. We are open to any further suggestions that the reviewer may have to improve the paper such that it can be accepted.

---

### Official Review · Reviewer_4uKr · 2023-11-02

**Soundness:** 2 fair
**Presentation:** 3 good
**Contribution:** 2 fair
**Rating:** 3
**Confidence:** 4

**Summary:**

This paper proposes a method called Molecular Conformer Fields (MCF) that parameterizes conformers as continuous functions mapping elements from the molecular graph to points in 3D space (3D coordinates). The problem is formulated as learning a distribution over these functions using a diffusion generative model (DDPM). MCF represents an advance in extending diffusion models to handle complex scientific problems in a scalable, simple, and effective manner. The backbone of the score field network is PerceiverIO, a transformer encoder-decoder architecture.

**Strengths:**

1. The paper is well organized and clearly written; and it’s easy to read.
2. The proposed method demonstrates noticeable enhancements on the small molecule dataset, specifically GEOM-QM9.

**Weaknesses:**

Inconsistency in Chemical Properties Experiment
In Section A.3.3, which pertains to the chemical properties experiment, this paper adopts experimental results from "Torsional Diffusion." However, there's a discrepancy in the subsets used. Specifically, the paper does not use the identical subset as "Torsional Diffusion," implying that they have selected different molecules. This raises concerns about the validity of their experimental comparison. For it to be persuasive, it should be more convincingly aligned.

Errors in Previous Studies
In section 1, page 2, For example, the quality of conformers from GeoMol (Ganea et al., 2021) and Torsional Diffusion (Jing et al., 2022) depends on the local substructure prediction model which is not differentiable.
In section 2, page 2, In GeoMol (Ganea et al., 2021), the authors propose a regression objective coupled with an Optimal Transport loss to predict the torsional angles of bonds that assemble substructures of a molecule

GeoMol's prediction of the local substructure is based on bond length, bond angle, and torsion angle. The definitions of torsion angle in GeoMol and Torsional Diffusion are different. For Torsional Diffusion, the torsion angle is based on rotatable bonds only.

Insufficient Performance
The GEOM-DRUGS dataset is considered a more challenging metric in this field. The method proposed in this paper demonstrates enhanced performance only regarding recall. Recall measures the capability to locate ground-truth conformers within the generated ones, and this metric is susceptible to influences from the training data. Notably, Torsional Diffusion depends on the local generated by RDKit, utilizing only 30 normalized conformers for each molecule during training. Although both the proposed method and Torsional Diffusion use the same dataset and split, the mean count of conformers in the GEOM-DRUGS dataset is approximately 100. This suggests that the boost in recall could result from the proposed method being trained on over 300% more conformers and having double the training epochs rather than the inherent efficacy of the method.

Furthermore, in terms of precision, the proposed method is worse than Torsional Diffusion, especially evident when Torsional Diffusion undergoes 50 denoising steps, as shown in [Table.9] of the supplemental, and MCF requires 1000 denoising steps.

High Computation Cost
The training settings for the proposed method use 8*A100 with a batch size of 64 (or one A100 with a batch size of 8). In contrast, Torsional Diffusion is trained on a singular A6000 with a batch size of 32. Consequently, the computational expense for each denoising phase for the proposed method is at least four times that of Torsional Diffusion. Considering Torsional Diffusion necessitates only 20 denoising steps, whereas MCF demands 1000, the proposed method, with a computational cost over two hundred times more, should ideally yield superior results, especially with larger molecules.

Absence of Runtime Experimental Results and Lack Significance
A pivotal metric for the conformer generation task is runtime. Even though some early deep-learning research might not have emphasized runtime, recent studies, such as GeoMol [Figure.7] and Torsional Diffusion [Table.2], regard it as a critical experimental metric. Historically, cheminformatics methods recognized runtime as a vital metric even before the advent of deep learning, as documented in [Conformation Generation: The State of the Art].

Furthermore, CREST, a method based on in metadynamics, is used to generate GEOM-Drugs dataset. The computational cost of CREST in GEOM-DRUGS is about 90 CPU core-hours per drug-like molecule, as detailed in [Torsional Diffusion Section 2, Page 2]. In contrast, Torsional Diffusion requires roughly 5-CPU core seconds for a conformer, while MCF might need between 15 to 20 CPU core-minutes (considering the computational cost is over 200 times).

Given that the average number (N) of conformers per molecule is about 100, MCF might need a minimum of 60 CPU core-hours to generate conformers (2N) for a single molecule. This speed is comparable to the dataset's generation rate (90 CPU core-hours per molecule). Hence, the proposed method's significance is low, especially when weighed against its high computational demands.

References:
Ganea, Octavian, et al. Geomol: Torsional geometric generation of molecular 3d conformer ensembles." Advances in Neural Information Processing Systems 34 (2021): 13757-13769.
Jing, B., Corso, G., Chang, J., Barzilay, R., & Jaakkola, T. (2022). Torsional diffusion for molecular conformer generation. Advances in Neural Information Processing Systems, 35, 24240-24253
Hawkins P. C. D. (2017). Conformation Generation: The State of the Art. Journal of chemical information and modeling, 57(8), 1747–1756.
Axelrod, S., Gómez-Bombarelli, R. GEOM, energy-annotated molecular conformations for property prediction and molecular generation. Sci Data 9, 185 (2022)
Pracht, P., , Bohle, F., , & Grimme, S., (2020). Automated exploration of the low-energy chemical space with fast quantum chemical methods. Physical chemistry chemical physics: PCCP, 22(14), 7169–7192. https://doi.org/10.1039/c9cp06869d

**Questions:**

For the chemical properties experiment, would it be possible to replicate the Torsional Diffusion experiment using your code and subset? This approach would ensure a more equitable comparison of results.

For the recall metric, could you consider training your proposed model using only 30 conformers for each molecule? Doing so could validate whether your method truly offers enhanced recall performance when using an identical dataset.

For runtime, are there any strategies you could employ to further reduce the computational cost, such as minimizing the number of denoising steps or downsizing the model? If the computational cost of your method is comparable to that of CREST, the approach generates the dataset, your proposed technique might lack significance.

---

> ### Author Response · Authors · 2023-11-15
> **Official Comments by Authors**
>
> We thank the reviewers for the constructive reviews. Please find below our detailed responses to the weaknesses and questions:
>
> 1. **Inconsistency in Chemical Properties Experiment In Section A.3.3, which pertains to the chemical properties experiment, this paper adopts experimental results from "Torsional Diffusion." However, there's a discrepancy in the subsets used. Specifically, the paper does not use the identical subset as "Torsional Diffusion," implying that they have selected different molecules. This raises concerns about the validity of their experimental comparison. For it to be persuasive, it should be more convincingly aligned.**
>    - We have added experimental results of MCF and reproduced Torsional Diffusion on same subset in Appendix A.3.3. It indicates that on the aligned subset, MCF generate conformers that are closer to the ground truth than Torsional Diffusion in terms of the ensemble properties. This further validates that MCF learns to model the distribution of physically/chemically valid molecular conformers.
>
> 2. **Errors in Previous Studies**
>    - We thank the reviewer for pointing out this. We have revised the description regarding GeoMol in section 1 and 2.
>
> 3. **Insufficient Performance The GEOM-DRUGS dataset is considered a more challenging metric in this field. The method proposed in this paper demonstrates enhanced performance only regarding recall. Recall measures the capability to locate ground-truth conformers within the generated ones, and this metric is susceptible to influences from the training data. Notably, Torsional Diffusion depends on the local generated by RDKit, utilizing only 30 normalized conformers for each molecule during training. Although both the proposed method and Torsional Diffusion use the same dataset and split, the mean count of conformers in the GEOM-DRUGS dataset is approximately 100. This suggests that the boost in recall could result from the proposed method being trained on over 300% more conformers and having double the training epochs rather than the inherent efficacy of the method.**
>    - We have updated the results on GEOM-DRUGS in Table 2, after training MCF with updated hyper-parameters (see Tab. 4). MCF now surpasses Torsional Diffusion on all metrics regarding recall and precision. On more challenging datasets like GEOM-DRUGS, MCF shows its advantage over other baselines.
>    - Also, we did not use the all the conformers in GEOM-DRUGS dataset during training. We follow the setting of GeoMol [1] and randomly pick at most 10 and 20 conformers for each molecule in training set of GEOM-QM9 and GEOM-DRUGS, respectively. Compared with 30 conformers per molecule in training Torsional Diffusion, our MCF is actually trained on less number of conformers. Trained on less data while achieving superior performance, MCF demonstrates the effectiveness in molecular conformer generation.
>
> 4. **Furthermore, in terms of precision, the proposed method is worse than Torsional Diffusion, especially evident when Torsional Diffusion undergoes 50 denoising steps and MCF requires 1000 denoising steps.**
>    - Following the reviewer’s suggestion, we have included experiments with DDIM sampling [2] with small number of timesteps in Appendix A.4 and Tab. 9. Our MCF demonstrates competitive performance using a reduced number of denoising steps. When using 20 denoising steps, which is the main setting Torsional Diffusion reported in the paper, MCF achieves better performance than Torsional Diffusion on the challenging GEOM-DRUGS dataset. Also, as shown in the Tab. 9 MCF surpasses Torsional Diffusion with 50 sampling steps. Indeed, Torsional Diffusion show advantage when the number of sampling steps is very small (3 or 5 steps). We attribute this to the fact that Torsional Diffusion uses a pre-processing step in which a cheimformatics method predicts local substructure in 3D spaces before applying their model. So even before inference the cheimformatics prediction provides a strong prior. In addition, we would like to mention there are lines of works concerning distillation of diffusion models [3] so that it can effectively sample in even one single step. Such techniques can be applied to MCF and further increase the sampling efficiency.
>
> ------- see following comments below -------

---

> > ### Author Response · Authors · 2023-11-15
> > **Official Comments by Authors (2)**
> >
> > ------- continued comments -------
> >
> > 5. **High Computation Cost**
> >    - MCF takes 0.5 GPU/s and 100 CPU/s to sample one conformer (much lower than CREST). As GPUs are widespread computing platforms in scientific computing, we believe GPU/s is a practical metric for assessing the inference time computational demands.  With MCF now greatly outperforming Torsional Diffusion in our updated results we provide an approach that is very efficient when running inference on GPU. When running inference on CPU, MCF provides better results with an increased runtime. It should also be noted that current runtime benchmark is conducted with batch size 1. In practice, one can stack multiple molecules as a batch and feed into MCF, which further improves the efficiency. In addition, there’s a few approaches one can use to distill MCF to use a very small number of denoising steps during inference [3,4,5] which will further decrease runtime.
> >
> > 6. **For the recall metric, could you consider training your proposed model using only 30 conformers for each molecule?**
> >    - As we explained in the previous response, we only use at most 20 conformers for each molecule when training on GEOM-DRUGS, which is a smaller training set than Torsional Diffusion. The updated Tab. 2 has shown that MCF achieved better performance than Torsional Diffusion on both precision and recall.
> >
> > 7. **For runtime, are there any strategies you could employ to further reduce the computational cost, such as minimizing the number of denoising steps or downsizing the model? If the computational cost of your method is comparable to that of CREST, the approach generates the dataset, your proposed technique might lack significance.**
> >    - In the response to previous questions, we have demonstrated efficient sampling technique, i.e., DDIM [2], can be directly applied to MCF while maintaining high conformer quality. There are also other efficient sampling methods like consistency model [4] that can be adapted as well. Besides, recent studies have proposed distillation methods for diffusion model that can achieve single-step diffusion with high quality [3,5]. There techniques can seamlessly be leveraged for MCF to increase sampling efficiency. We have also included more discussions about possible strategies to further reduce the computational cost in Appendix A.1.
> >
> > references:
> >
> > [1] Ganea, O., Pattanaik, L., Coley, C., Barzilay, R., Jensen, K., Green, W. and Jaakkola, T., 2021. Geomol: Torsional geometric generation of molecular 3d conformer ensembles. Advances in Neural Information Processing Systems, 34, pp.13757-13769.
> >
> > [2] Song, J., Meng, C. and Ermon, S., 2021. Denoising diffusion implicit models. In International Conference on Learning Representations.
> >
> > [3] Gu, J., Zhai, S., Zhang, Y., Liu, L. and Susskind, J.M., 2023, July. Boot: Data-free distillation of denoising diffusion models with bootstrapping. In ICML 2023 Workshop on Structured Probabilistic Inference & Generative Modeling.
> >
> > [4] Song, Y., Dhariwal, P., Chen, M. and Sutskever, I., 2023. Consistency models.
> >
> > [5] Berthelot, D., Autef, A., Lin, J., Yap, D.A., Zhai, S., Hu, S., Zheng, D., Talbot, W. and Gu, E., 2023. TRACT: Denoising Diffusion Models with Transitive Closure Time-Distillation. arXiv preprint arXiv:2303.04248.

---

### Official Review · Reviewer_Y3o3 · 2023-11-04

**Soundness:** 2 fair
**Presentation:** 1 poor
**Contribution:** 2 fair
**Rating:** 3
**Confidence:** 4

**Summary:**

the paper propose diffusion model to generate molecule conformation based on atom-to-atom graphs. the paper can generate a distribution of confirmations.

the formulation of the framework of this paper is not clear. I will elaborate in later section.

I checked out Jing et al 2022 for the definition of average minimum RMSD. indeed Jing does not describe this method, the authors should cite the paper Jing cited, that is: Ganea et al. GEOMOL: Torsional Geometric Generation of Molecular 3D Conformer Ensembles.

In Ganea et al, AMR is given in Eq.(5). there is also COV in Eq.(5) which describes the coverage. Coverage would be a more important measure.

**Strengths:**

the method can generate a distribution of conformations rather than just generate one confirmation. this has good advantage because molecules in the real world takes a distribution of confirmation following the Boltzmann distribution and obeys the law of thermodynamics.

**Weaknesses:**

if we have n atoms in one molecule, then the conformation space is \mathbb{R}^{3n} instead of \mathbb{R}^3. in the paper authors indicate f: G -> \mathbb{R}^3. I am confused.

section 4.1. Score field network: the author cite and use perceiverIO net and explain why they use that. this is good. the author should explain further for the benefits of the readers, what line 8 of algorithm 1 entails. \epsilon_q ~ N(0,I), line 8 want \epsilon_\theta to map to N(0,I). what is the physical significance in the context of molecule conformations?

the authors claimed that the strength of their method is that there is no need to know some domain knowledge such as torsion angles of molecules. in my opinion this is a serious weakness. would the predictions generate physically non-viable confirmations if it disregard some basic physics and chemistry?

since diffusion model will generate a distribution of conformations, does the distribution follows the distribution of statistical physics? e.g. Boltzmann distribution.

molecules are rotational invariant. that means, one can rotate the conformation by any angles in 3D and the conformation remains a valid conformation. is the diffusion model able to generated rotational invariant distribution of conformation?

**Questions:**

page 7, paragraph 1, "We generate 2K confirmers for a molecule with K ground truth conformers", how does the author do this? it is important to have 'correct' ground truth. if the authors use MD or MCMC with proper settings, then the ground truth can be correct. what are the settings? explicit water? mean field? what are the molecular interactions?

does the ground truth distribution obeys statistical physics law? NVT? grand canonical ensemble? micro-canonical ensemble?

---

> ### Author Response · Authors · 2023-11-15
> **Official Comments by Authors**
>
> We thank the reviewers for the constructive reviews. Please find below our detailed responses to the weaknesses and questions:
>
> 1. **If we have n atoms in one molecule, then the conformation space is $\mathbb{R}^{3n}$ instead of $\mathbb{R}^3$. in the paper authors indicate $f: G → \mathbb{R}^3$.**
>    - Here, the notation represents a field (or implicit function) that maps any node from a graph to 3D Euclidean space. Namely it means mapping an atom (node) in a molecular graph to its 3D coordinates in the Euclidean space instead of mapping a molecular graph to a 3D conformer.
>
> 2. **Section 4.1. Score field network: the author cite and use perceiverIO net and explain why they use that. this is good. the author should explain further for the benefits of the readers, what line 8 of algorithm 1 entails. \epsilon_q ~ N(0,I), line 8 want \epsilon_\theta to map to N(0,I). what is the physical significance in the context of molecule conformations?**
>    - The loss in line 8 of algorithm 1 is adapted from DDPM loss. $\epsilon_q$ are sampled Gaussian noises that are injected to corrupt the output space of conformer field (i.e, 3D atomic coordinates). We train the score network $\epsilon_\theta$ to predict the Gaussian noise added to the coordinates. Therefore, the gradient is not intended to push $\epsilon_\theta$ to a standard Gaussian, but rather predict the sampled noise $\epsilon_q$ given the corrupted context and query pairs at current timestep $t$. So that in inference, MCF can generate realistic molecular conformers starting from random Gaussian noise.
>
> 3. **The authors claimed that the strength of their method is that there is no need to know some domain knowledge such as torsion angles of molecules. in my opinion this is a serious weakness. would the predictions generate physically non-viable confirmations if it disregard some basic physics and chemistry?**
>    - We updated Tab. 2 with results from an updated MCF model, which demonstrates superior performance over all previous approaches on the challenging GEOM-DRUGS dataset. Also, in updated Tab. 8, MCF achieves competitive performance on ensemble property prediction. We note that although MCF does not enforce explicit inductive biases (eg. modeling torsional angles) it generates physically/chemically realistic samples. In addition other machine learning domains like computer vision [1,2] and natural language processing [3,4] that attention-based transformers and its variants can generalize well to various downstream applications when pre-trained on sufficiently large datasets. This indicates that if the domain knowledge is properly included in sufficient training data (e.g., the molecular conformers with low energies investigated in this work), the model can implicitly learn inductive biases from data without the need to explicitly enforcing them into the architecture. Besides, it should be noted that integrating domain knowledge also comes at a cost. For example, in Torsional Diffusion, one needs to use rule-based method to find the rotatable bonds which can fail in some cases especially for large complex molecules. It also relies on cheminformatic methods to predict local substructures which may neglect long-range interactions between substructures and restrict the conformer quality by the local methods. On the other hand, MCF provides a framework that directly operates on Euclidean coordinates which is easier to scale up both in terms of number of parameters as well as larger molecular systems.
>
> 4. **Since diffusion model will generate a distribution of conformations, does the distribution follows the distribution of statistical physics? e.g. Boltzmann distribution.**
>     Diffusion generative models are trained to model an empirical distribution of training data. Following the setting of previous work [5], we randomly pick at most 10 and 20 conformers for each molecule during training for GEOM-QM9 and GEOM-DRUGS datasets [6], respectively. Therefore, the training data doesn’t follow the Boltzmann distribution and our current MCF does not exactly generate Boltzmann distributed conformers. However, if provided training data that follows Boltzmann distribution, diffusion models like MCF are expected learn to the model the distribution as shown in some recent works [7]. Also, we can apply weighted sampling during training if Boltzmann weights are available for training conformers, which can force generating conformers in Boltzmann distribution. Another extension could be integrating flow matching [8] framework which provides exact estimation of log-likelihood. Flow matching provides the flexibility to map between arbitrary distributions. Therefore, instead of sampling from a standard Gaussian distribution, one may start from a Boltzmann distribution in inference. We have added discussions to Appendix A.1 as future works to extend MCF for generating conformers following Boltzmann distributions.
>
> ------- see following comments below -------

---

> > ### Author Response · Authors · 2023-11-15
> > **Official Comments by Authors (2)**
> >
> > ------- continued comments -------
> >
> > 5. **Molecules are rotational invariant. that means, one can rotate the conformation by any angles in 3D and the conformation remains a valid conformation. is the diffusion model able to generated rotational invariant distribution of conformation?**
> >    - In this work, we highlight the question of whether in-/equi-variant neural networks are required for models that generate conformers. In particular, we note that what matters in conformer generation is the intrinsic geometry of the conformer and not the rigid coordinate system (as reflected by the metrics used in to evaluate approaches, which involve an alignment step, thus disregarding information about rigid SO(3) transformations). Using MCF, one can always sample conformers from the model and then apply a post-hoc SO(3) transformation if needed.
> >
> > 6. **Page 7, paragraph 1, "We generate 2K confirmers for a molecule with K ground truth conformers", how does the author do this? it is important to have 'correct' ground truth. if the authors use MD or MCMC with proper settings, then the ground truth can be correct. what are the settings? explicit water? mean field? what are the molecular interactions?**
> >    - Following the previous works [5,9], we use test data from standard GEOM-QM9 and GEOM-DRUGS benchmarks as ground truth. GEOM datasets applies semi-empirical DFT to calculate the molecular conformers. Each molecule can contain different number of conformers. In testing, we use all the K conformers for a molecule and sample 2K conformers to calculates the coverage and AMR metrics as [5,9] so that we can collect numbers for apple-to-apple comparison. Also, when trained with other data from MD or MCMC, our method can be applied to model the distributions as well.
> >
> > 7. **Does the ground truth distribution obeys statistical physics law? NVT? grand canonical ensemble? micro-canonical ensemble?**
> >     - We follow the setting in GeoMol to randomly pick at most 10 or 20 conformers for each molecule in training on GEOM-QM9 and GEOM-DRUGS, respectively. Therefore, the ground truth training data doesn’t exactly obey statistical physics laws. Note that this is done in order to provide a fair comparison with previous work on conformer generation.
> >
> >
> > references:
> >
> > [1] He, K., Chen, X., Xie, S., Li, Y., Dollár, P. and Girshick, R., 2022. Masked autoencoders are scalable vision learners. In Proceedings of the IEEE/CVF conference on computer vision and pattern recognition (pp. 16000-16009).
> >
> > [2] Kirillov, A., Mintun, E., Ravi, N., Mao, H., Rolland, C., Gustafson, L., Xiao, T., Whitehead, S., Berg, A.C., Lo, W.Y. and Dollár, P., 2023. Segment anything. arXiv preprint arXiv:2304.02643.
> >
> > [3] Devlin, J., Chang, M.W., Lee, K. and Toutanova, K., 2018. Bert: Pre-training of deep bidirectional transformers for language understanding. arXiv preprint arXiv:1810.04805.
> >
> > [4] Brown, T., Mann, B., Ryder, N., Subbiah, M., Kaplan, J.D., Dhariwal, P., Neelakantan, A., Shyam, P., Sastry, G., Askell, A. and Agarwal, S., 2020. Language models are few-shot learners. Advances in neural information processing systems, 33, pp.1877-1901.
> >
> > [5] Ganea, O., Pattanaik, L., Coley, C., Barzilay, R., Jensen, K., Green, W. and Jaakkola, T., 2021. Geomol: Torsional geometric generation of molecular 3d conformer ensembles. Advances in Neural Information Processing Systems, 34, pp.13757-13769.
> >
> > [6] Axelrod, S. and Gomez-Bombarelli, R., 2022. GEOM, energy-annotated molecular conformations for property prediction and molecular generation. Scientific Data, 9(1), p.185.
> >
> > [7] Arts, M., Garcia Satorras, V., Huang, C.W., Zügner, D., Federici, M., Clementi, C., Noé, F., Pinsler, R. and van den Berg, R., 2023. Two for one: Diffusion models and force fields for coarse-grained molecular dynamics. Journal of Chemical Theory and Computation, 19(18), pp.6151-6159.
> >
> > [8] Lipman, Y., Chen, R.T., Ben-Hamu, H., Nickel, M. and Le, M., 2022. Flow matching for generative modeling. arXiv preprint arXiv:2210.02747.
> >
> > [9] Jing, B., Corso, G., Chang, J., Barzilay, R. and Jaakkola, T., 2022. Torsional diffusion for molecular conformer generation. Advances in Neural Information Processing Systems, 35, pp.24240-24253.

---

> > > ### Comment · Reviewer_Y3o3 · 2023-11-22
> > > **keep score**
> > >
> > > I like to thank the authors for their effort. this is a nice piece of work. I encourage the authors to consider all comments and improve their paper for future submissions. I like to keep my scores.

---

> > > > ### Author Response · Authors · 2023-11-22
> > > >
> > > > We thank the reviewer for engaging with us and for their feedback, which we have carefully addressed in the revision of the manuscript. We kindly ask the reviewer to point out what they think we missed in our response. We believe we've fully addressed all the reviewers suggestions.

---

### Official Review · Reviewer_QmU2 · 2023-11-13

**Soundness:** 2 fair
**Presentation:** 3 good
**Contribution:** 2 fair
**Rating:** 5
**Confidence:** 4

**Summary:**

This paper proposes a generative filed model to generate a 3D conformer based on the 2D molecular graph. The main contribution of this paper is the extension of the diffusion probabilistic field to the problem of generation conformers. The proposed Molecular Conformer Field (MCF) is defined from eigenvectors of the 2D molecular graphs to 3D atomic positions. Experimental results show the proposed model outperforms the baselines.

**Strengths:**

S1: The studied problem is interesting and useful.

S2: It is interesting to apply the diffusion probabilistic field to the problem of generation conformers.

S3: The paper is well-written.

**Weaknesses:**

W1: The novelty is limited from a technological perspective. I see little difference between the diffusion probabilistic field and the proposed molecular conformer field.

W2: Using eigenvectors of the position of the graph in the field seems dangerous for the folloing two reasons. (1) This modeling does not take key information, such as atomic numbers, into consideration. (2) The eigenvector based position cannot effectively indicate distance in the graph without the corresponding eigenvalues.

**Questions:**

Q1: Can this proposed method distinguish two different 2D molecular graphs that have the same eigenvectors but diffenrent eigenvalues?


After read the authors' response, I increase my score to "5: marginally below the acceptance threshold".

---

> ### Author Response · Authors · 2023-11-15
> **Official Comments by Authors**
>
> We thank the reviewer for the constructive reviews. Please find below our detailed responses to the concerns:
>
> 1. **The novelty is limited from a technological perspective. I see little difference between the diffusion probabilistic field and the proposed molecular conformer field.**
>    - In this work we generalize the problem setting in DPF to deal with functions on graphs (as opposed to functions on Euclidean spaces). Our proposed approach is conceptually simple to implement while being very powerful in its application. **Note that MCF sets the current state-of-the-art for conformer generation.** We believe that approaches that are both simple to implement, scalable and powerful in their empirical results tend to have the longest lasting impact in the field (ref the bitter lesson by Rich Sutton http://www.incompleteideas.net/IncIdeas/BitterLesson.html) .
>
> 2. **Using eigenvectors of the position of the graph in the field seems dangerous for the following two reasons. (1) This modeling does not take key information, such as atomic numbers, into consideration. (2) The eigenvector based position cannot effectively indicate distance in the graph without the corresponding eigenvalues.**
>    - In MCF, we use atomic features which are concatenated with the graph Laplacians in both query and context pairs, including atomic numbers, aromatic, degree, hybridization, formal charge, etc. Therefore MCF has included essential atomic information that models the molecules. We have added an extra section A.2.2 in Appendix to clarify the atomic features used in this work. Though we didn’t include eigenvalues as the input to MCF, the eigenvectors are ranked based on eigenvalues. This can provide the extra information regarding the distance and connectivity between atoms. Finally, in practice we see that even without eigenvalues MCF sets the new state-of-the-art for conformer generation.
>
> 3. **Can this proposed method distinguish two different 2D molecular graphs that have the same eigenvectors but different eigenvalues?**
>    - As mentioned in response to the previous question, eigenvectors are ranked based on eigenvalues. Therefore,  MCF can still distinguish graphs with same eigenvectors but different eigenvalues. In addition, we use atomic features as input to MCF which solves this type of problem in practice.

---

> > ### Comment · Reviewer_QmU2 · 2023-11-20
> > **Thanks for your reply**
> >
> > The response partially addressed my concerns. I would like to increase my score to  "5: marginally below the acceptance threshold"

---

> > > ### Author Response · Authors · 2023-11-20
> > > **Official Comments by Authors**
> > >
> > > We thank the reviewer for the increased score and acknowledging our contributions. As the reviewer mentioned that our response partially addressed the concerns. We kindly request for more details about the remaining issues that haven't been fully answered in our previous response. We believe it will allow us to further improve the quality of the manuscript and ensure all your concerns are thoroughly considered in the revision. We appreciate your time and look forward to your timely reply.

---

### Author Response · Authors · 2023-11-15
**To All Reviewers and ACs**

Dear Reviewers and ACs:

We thank all reviewers for their valuable comments. We have include detailed point-by-point response to all the questions and highlighted revisions in the updated manuscript. Here we list a few major updates that address the shared concerns from reviewers.

1. We trained an MCF model with updated hyper-parameters (eg. larger embedding dimension and more epochs, see Tab. 4). The new model has achieved state-of-the-art performance on the challenging GEOM-DRUGS and GEOM-XL datasets. We show the updated Tab. 2 & 3 below. This demonstrates the effectiveness of MCF in solving molecular conformer generation and further showcases the flexibility of scaling up the framework for better performance.

Molecule conformer generation results on GEOM-DRUGS:

|| COV-R mean | COV-R median | AMR-R mean | AMR-R median | COV-P mean | COV-P median | AMR-P mean | AMR-P median |
|---|---|---|---|---|---|---|---|---|
| GeoDiff         | 42.1 | 37.8 | 0.835 | 0.809 | 24.9 | 14.5 | 1.136 | 1.090 |
| GeoMol          | 44.6 | 41.4 | 0.875 | 0.834 | 43.0 | 36.4 | 0.928 | 0.841 |
| Torsional Diff. | 72.7 | 80.0 | 0.582 | 0.565 | 55.2 | 56.9 | 0.778 | 0.729 |
| MCF (ours)      | **81.6** | **89.2** | **0.468** | **0.438** | **61.6** | **62.5** | **0.705** | **0.650** |

Molecule conformer generation results on GEOM-XL:

|| AMR-R mean | AMR-R median | AMR-P mean | AMR-P median | # mols |
|---|---|---|---|---|---|
| GeoDiff                    | 2.92 | 2.62 | 3.35 | 3.15 | - |
| GeoMol                     | 2.47 | 2.39 | 3.30 | 3.14 | - |
| Torsional Diff.            | 2.05 | 1.86 | **2.94** | 2.78 | - |
| MCF (ours)                 | **2.04** | **1.59** | 2.97 | **2.51** |102|
| Torsional Diff. (our eval) | 1.93 | 1.86 | 2.84 | 2.71 |77 |
| MCF (ours)                 | **1.77** | **1.59** | **2.62** | **2.37** |77 |

2. MCF can naturally be integrated with efficient sampling techniques to significantly decrease the sampling steps while maintaining high-quality sampling performance. We have added the Tab. 9 (also shown below) to the Appendix.

|| steps | COV-R mean | COV-R median | AMR-R mean | AMR-R median | COV-P mean | COV-P median | AMR-P mean | AMR-P median |
|---|---|---|---|---|---|---|---|---|---|
| Torsional Diff. | 3  | 42.9 | 33.8 | 0.820 | 0.821 | 24.1 | 11.1 | 1.116 | 1.100 |
| Torsional Diff. | 5  | 58.9 | 63.6 | 0.698 | 0.685 | 35.8 | 26.6 | 0.979 | 0.963 |
| Torsional Diff. | 10 | 70.6 | 78.8 | 0.600 | 0.580 | 50.2 | 48.3 | 0.827 | 0.791 |
| Torsional Diff. | 20 | 72.7 | 80.0 | 0.582 | 0.565 | 55.2 | 56.9 | 0.778 | 0.729 |
| Torsional Diff. | 50 | 73.1 | 80.4 | 0.578 | 0.557 | 57.6 | 60.7 | 0.753 | 0.699 |
| MCF (DDIM)      | 3  | 15.05 | 0.00 | 1.032 | 1.041 | 5.71 | 0.00 | 1.521 | 1.525 |
| MCF (DDIM)      | 5  | 42.86 | 35.50 | 0.813 | 0.814 | 20.07 | 11.54 | 1.149 | 1.149 |
| MCF (DDIM)      | 10 | 74.14 | 83.25 | 0.610 | 0.601 | 48.95 | 46.35 | 0.841 | 0.813 |
| MCF (DDIM)      | 20 | 80.87 | 88.89 | 0.522 | 0.504 | 59.72 | 60.23 | 0.745 | 0.708 |
| MCF (DDIM)      | 50 | 81.87 | 88.89 | 0.481 | 0.459 | 62.01 | 62.53 | 0.708 | 0.660 |
| MCF (DDIM)      |100 | 81.97 | 89.10 | 0.466 | 0.441 | 62.81 | 63.64 | 0.693 | 0.641 |
| MCF (DDPM)      |1000| 81.62 | 89.22 | 0.468 | 0.438 | 61.63 | 62.50 | 0.705 | 0.650 |

3. We have added experiments of ensemble property evaluation using the same test set for Torsional Diffusion and our proposed MCF in Appendix A.3.3 and Tab. 8.


We believe that the quality of the paper has been substantially improved through the revision. Please do not hesitate to contact us if you have any further concerns that we can address before you make decisions on changing your evaluation scores.


Thanks,

Authors

---

### Author Response · Authors · 2023-11-21
**Official Comment by Authors**

Dear reviewers,

Thanks for your time and commitment reviewing our submission, we believe your comments have helped improve the quality of the paper! We have addressed reviewers comments and highlighted changes in the manuscript in red color.

Since we are into the last two days of the discussion phase, we are eagerly looking forward to your post-rebuttal responses. Please do let us know if there are any additional suggestions to improve the paper such that it can be accepted.

Authors

---

### Meta-Review · Area_Chair_RUH6 · 2023-12-05

**Metareview:**

The paper presents Molecular Conformer Fields (MCF), a novel approach to generating 3D molecular conformers from molecular graphs. MCF models these conformers as continuous functions that map graph elements to points in 3D space, employing a diffusion generative model for learning a distribution over these functions.

**Justification For Why Not Higher Score:**

Main weaknesses pointed out by the reviewers include limited novelty, with results not showing substantial improvement over existing works. Additionally, the approach has a high computational cost. Another concern is that MCF does not incorporate inductive biases, which makes it less efficient in terms of sample usage. This lack of inductive biases could potentially limit the method's efficiency and effectiveness in practical applications.

**Justification For Why Not Lower Score:**

N/A

---

### Decision · Program_Chairs · 2024-01-16

Reject